# Targeting Metabolism: Innovative Therapies for MASLD Unveiled

**DOI:** 10.3390/ijms26094077

**Published:** 2025-04-25

**Authors:** Weixin Wang, Xin Gao, Wentong Niu, Jinping Yin, Kan He

**Affiliations:** 1Department of Pharmacology, College of Basic Medical Sciences, Jilin University, Changchun 130021, China; wangwx8221@mails.jlu.edu.cn (W.W.); niuwt8222@mails.jlu.edu.cn (W.N.); 2School of Public Health, Jilin University, Changchun 130021, China; gaoxin2723@mails.jlu.edu.cn; 3NHC Key Laboratory of Radiobiology, School of Public Health, Jilin University, Changchun 130041, China; yinjp9921@mails.jlu.edu.cn

**Keywords:** MASLD, metabolic syndrome, targeted therapy, lipidosis, inflammatory response

## Abstract

The recent introduction of the term metabolic-dysfunction-associated steatotic liver disease (MASLD) has highlighted the critical role of metabolism in the disease’s pathophysiology. This innovative nomenclature signifies a shift from the previous designation of non-alcoholic fatty liver disease (NAFLD), emphasizing the condition’s progressive nature. Simultaneously, MASLD has become one of the most prevalent liver diseases worldwide, highlighting the urgent need for research to elucidate its etiology and develop effective treatment strategies. This review examines and delineates the revised definition of MASLD, exploring its epidemiology and the pathological changes occurring at various stages of the disease. Additionally, it identifies metabolically relevant targets within MASLD and provides a summary of the latest metabolically targeted drugs under development, including those in clinical and some preclinical stages. The review finishes with a look ahead to the future of targeted therapy for MASLD, with the goal of summarizing and providing fresh ideas and insights.

## 1. Introduction

The increasing global prevalence of metabolic syndrome, including obesity and type 2 diabetes mellitus (T2DM), is significantly contributing to the rise in chronic hepatic steatosis associated with these metabolic disorders [1,2,3,4,5,6]. Current estimates indicate that the global prevalence of fatty liver disease secondary to these metabolic factors ranges from 25% to 30%, making it one of the most common chronic liver diseases globally. The progression of this condition to more advanced stages, such as steatohepatitis and cirrhosis, is associated with malignant injury resulting in liver function impairment and the development of hepatocellular carcinoma (HCC) that can lead to severe outcomes, including liver failure, and even primary malignant HCC in advanced stages. These complications can significantly diminish patient survival rates [4,7,8]. To enhance the planning of interventions addressing the root causes and treatment of the disease, a term for the reclassification of hepatic steatosis, metabolic-dysfunction-associated steatotic liver disease (MASLD), was officially introduced by the American Association for the Study of Liver Diseases (AASLD), the European Association for the Study of the Liver (EASL), and the Asociación Latinoamericana de Estudio del Hígado (ALEH) at the EASL 2023 conference [9]. The EASL-EASD-EASO Clinical Practice Management Guidelines 2024 subsequently offer a comprehensive definition of MASLD, along with pertinent diagnostic criteria and treatment guidelines [9]. Unlike NAFLD, which focuses on fatty liver disease resulting from non-alcoholic factors, MASLD highlights the interplay between systemic metabolism and the progression of the disease. This advanced understanding is crucial for enabling more accurate diagnostic and therapeutic approaches, thereby fostering the development of innovative frameworks and concepts that will shape the future management of liver diseases.

As a result, given the emphasis on ‘metabolic abnormality’ and the ‘continuous progression of illness spectrum’ in MASLD, it is critical that we develop a system of metabolism-related targeted therapeutic medications based on the disease status of each stage of MASLD. This review focuses on the revised definition of MASLD, discusses the changes and benefits of the reclassification, and examines the disease’s etiology across its range. It uses the progressive stages of disease progression and exacerbation in the MASLD spectrum as the central story premise.

The review also discusses the abnormalities in metabolism associated with each stage and provides a more thorough introduction to the metabolism-related targets of MASLD. Finally, it summarizes and analyzes these targeted therapeutic agents. This serves as a resource for both the therapeutic treatment of patients at various phases of MASLD and future research in the field of MASLD treatment.

Furthermore, the direction of targeting research reflected in many molecular investigations is also included in this study. To sum up, this study offers viewpoints and insights into upcoming therapeutic approaches, as well as the investigation and creation of tailored medications in the field of MASLD. Future developments in the field of molecular targeted therapeutics are expected to help MASLD patients more effectively, alleviate their condition, and enhance their quality of life. A list of promising novel targets that could be used as a consultation for upcoming therapeutic research is also included in the review.

## 2. The New Classification: MASLD

Due to the constraints of the nomenclature for non-alcoholic fatty liver disease (NAFLD), and in order to enhance etiological planning and treatment program design, the International Consensus Panel on Fatty Liver Disease (ICPFLD) introduced the term metabolic-associated fatty liver disease (MAFLD) in 2020 [10]. This new phrase underscores the pivotal role of metabolic disruption in disease progression [1,11,12]. In 2023, the European Association for the Study of the Liver (EASL) officially introduced the term MASLD following extensive research on liver diseases, aimed at more precisely characterizing the condition, particularly the connection between fatty liver disease and metabolic disorders. MASLD is characterized by excess triglyceride accumulation in the liver alongside at least one cardiometabolic risk factor and has been incorporated into the updated consensus definition of steatotic liver disease (SLD) [9]. The 2024 EASL-EASD-EASO Clinical Practice Guidelines on the Management of Metabolic Dysfunction-Associated Steatotic Liver Disease (MASLD) offers a thorough summary of the revisions concerning the definition, prevention, screening, diagnosis, and treatment of MASLD [9]. Notably, in addition to MASLD, the management consensus states that SLD also comprises MASLD with a moderate alcohol intake (MetALD), alcoholic liver disease (ALD), cause-specific fatty liver disease (e.g., monogenic, and pharmacologic illnesses), and cryptogenic fatty liver disease. Of these, patients with MASLD who also consume moderate amounts of alcohol (140–350 g per week for women and 210–420 g per week for males) are particularly referred to as having MetALD, a subtype of MASLD. Since alcohol use dramatically raises the risk of liver fibrosis and hepatocellular carcinoma, this classification attempts to differentiate the superimposed effect of alcohol on the development of metabolic liver disease, elucidating the metabolic factors’ and alcohol’s combined pathogenic effect. The diagnosis of MASLD or MetALD is based on whether alcohol is consumed or not, as well as if alcohol consumption is excessive (≥20 g/day for women and ≥30 g/day for males).

The name MASLD provides a reclassification of fat-related liver disease, highlighting the connection between fatty liver and metabolic dysfunction, rather than just attributing it to non-alcoholic causes. It reclassifies a range of diseases according to metabolic dysfunction: steatosis, metabolic-dysfunction-associated steatohepatitis (MASH), fibrosis, cirrhosis, and MASH-related HCC [9,13].

The reclassified term has had a profound impact on the field of medical research, particularly in terms of advancements in the diagnosis and treatment of disease. The reclassified definition broadens the parameters of screening and enables the early identification of a greater number of individuals at risk for metabolic disorders, including those who are overweight or obese, those diagnosed with type 2 diabetes, and other high-risk populations. Individuals may be assessed for MASLD using fundamental liver function tests and ultrasound imaging, even in the absence of overt liver disease manifestations. MASLD recategorizes diseases according to metabolic disorders, hence improving the focus on metabolically relevant causes that lead to the development of MASLD in patients, alongside traditional treatment methods. The revised definition establishes a framework for risk stratification and therapeutic therapy, including both non-pharmacological and pharmacological interventions, with distinct protocols for individuals at different stages of MASLD.

Moreover, the revised criteria enable the distinction among several subtypes of fatty liver disease, each potentially requiring unique therapy approaches. Thus, it is expected that the definition will foster a change in understanding among patients and healthcare providers concerning the severity of the disease and the importance of metabolic risk factors. This, consequently, should yield enhanced disease prevention and management [9].

## 3. The Contemporary Epidemiological Situation

The enhancement in nutritional standards, alteration of dietary composition, and rise in hazardous eating practices in recent years have led to a gradual escalation in the incidence of metabolic illnesses within the body. As a result, organ abnormalities resulting from these illnesses are increasingly common [14,15]. This phenomenon is particularly pronounced among individuals with underlying health conditions, such as obesity (BMI ≥ 25 kg/m^2^ is considered overweight, and BMI ≥ 30 kg/m^2^ is considered obese) [16,17], type 2 diabetes (FPG ≥ 7.0 mmol/L, 2 hPG ≥ 11.1 mmol/L, or HbA1c ≥ 6.5%) [18,19,20,21,22], dyslipidemia (e.g., serum triglyceride ≥ 1.7 mmol/L), and so on. Insulin resistance [23,24], fat metabolism disorders [25], and abnormalities of the intestinal flora [26] have been identified as contributing to the metabolic dysregulation of the liver. The accumulation of fat in the liver has also been demonstrated [25], as has the promotion of an inflammatory response [27]. These factors have demonstrated a cumulative impact on the normal functioning of the liver. The liver’s health is subsequently impacted by these metabolic factors [28,29,30,31].

With an estimated frequency of 25–30% in the world’s population, it is, therefore, regarded as one of the most common chronic liver disorders worldwide [6]. This suggests that MASLD may impact roughly one in four to five people globally, with prevalence rates showing a steadily increasing trend [6,32,33]. A comparable trend is evident in China, with a rapid increase in the prevalence of MASLD [14,34,35]. The prevalence of MASLD among Chinese adults is approximately 20–30%, aligning with global figures, and it has become one of the most common chronic progressive liver diseases in China. The onset of the condition is increasingly occurring at younger ages, affecting not only adults but also children and adolescents; the global frequency among children and adolescents is approximately 10% [6,36]. Childhood obesity constitutes a major risk factor for the onset of MASLD in pediatric populations [37]. The prevalence of MASLD in obese children can range from 30 to 50 percent.

## 4. Pathological Changes in MASLD

This section outlines the pathological alterations associated with MASLD. The EASL-EASD-EASO Clinical Practice Management Guidelines for MASLD classify the pathological spectrum of MASLD into categories ranging from mild to severe as follows [9]: fatty liver (steatosis); metabolic-dysfunction-associated steatohepatitis (MASH); fibrosis and cirrhosis; and MASH-related HCC. It is worth noting that this spectrum represents a nonlinear relationship; specifically, while MASLD suggests a theoretical progression of exacerbation, some patients may display fibrosis without progressing to MASH, and others may even revert from severe MASH to steatosis.

The specific pathophysiological changes that occur are as follows (Figure 1):

### 4.1. Fatty Liver

Hepatic steatosis is the hallmark of MASLD; this stage represents the preliminary phase of MASLD, characterized primarily by disorders in lipoprotein metabolism resulting from impaired lipid uptake by hepatocytes, enhanced fatty acid synthesis, and reduced fatty acid oxidation [38,39,40]. Thus, lipid buildup and hepatic steatosis occur [41]. Hepatocyte fat droplets, mostly macrovesicular steatosis, are the main pathogenic change. Macrovesicular steatosis is caused by enormous lipid droplets in hepatocytes that push the nucleus to the cell’s periphery. The damaged hepatocyte grows and becomes vacuolated. Macrovesicular steatosis ranges from modest, with a few lipid droplets in individual hepatocytes, to severe, involving numerous [18,42]. During steatosis, fat buildup causes liver enlargement, resulting in a blunt, sticky texture and oily-yellow appearance.

### 4.2. MASH

MASH is characterized by the histological features of hepatocellular ballooning and lobular inflammation. This stage involves hepatic-steatosis-related inflammation and cellular damage. MASLD progresses from a simple lipid buildup to MASH due to increased liver inflammation, oxidative stress, and hepatocyte apoptosis and necrosis [27,39,43,44,45,46]. Lymphocytes, monocytes, and neutrophils infiltrate liver tissue. Hepatocellular macrovesicular steatosis following damage is slow. Fat buildup damages hepatocyte organelles, causing ballooning changes [47]. This alters the intracellular osmotic pressure and causes significant water inflow; endoplasmic reticulum pools dilate, mitochondrial cristae decrease, and the organelle structure becomes disorganized [48]. This causes the hepatocytes, which have balloon-shaped, loose, hyaline cytoplasm, to grow [41]. Additionally, this process causes liver apoptosis, which manifests as hepatocyte wrinkles, nuclear chromatin condensation, and marginalization, resulting in apoptotic vesicles [40,41,49]. This could result in necrosis, which causes a stronger inflammatory response in the tissue around the necrotic hepatocytes. With this process, hepatocyte membranes burst, spilling cell contents into the extracellular space. Leaking cytoplasmic elements include enzymes and organelles. Hepatocyte oedema and inflammation cause the liver to swell macroscopically, making it fragile and without sharp edges. A palpable nodule and oily feeling may indicate advanced stages.

### 4.3. Liver Fibrosis and Cirrhosis

Hepatic fibrosis is a significant indication of MASLD progression. Chronic liver stimulation by fat accumulation, inflammation, and other adverse factors activates hepatic stellate cells (HSCs). These cells become myofibroblast-like and produce large amounts of collagen and fibronectin [41,47,50]. The gradual deposition of the extracellular matrix (ECM) causes fibrous septa in the liver [51,52]. ‘Perisinusoidal fibrosis’ or ‘periportal fibrosis’ are the early stages, where the fibrous septa are primarily found surrounding the central vein or on the outskirts of hepatic lobules [51,53]. As cirrhosis progresses, fibrous septa widen and join, dividing the hepatic lobules into irregular pseudo-lobules. Due to hepatocyte disintegration in the pseudolobules, the central vein may be absent, deformed, or multiple [41]. The fibrous tissue compresses and twists the portal vein and hepatic vein branches, narrowing and occluding them and causing portal hypertension [54,55]. Anomalous anastomotic branches connecting hepatic arteries and veins can worsen blood circulation [56]. Fibrous tissue growth makes the liver hard and uneven; sectioning the liver produces masses and greyish-white fibrous septa. Cholestasis and hepatocyte steatosis can tint the liver yellowish-brown or green. Advanced fibrotic livers are smaller and uneven. The tissue disturbance reduces hepatic protein production and detoxification [57,58].

### 4.4. MASH-Associated HCC

MASH, cirrhosis, hepatotoxic causes, and repeated injury may cause linked cancers, including HCC [59,60,61,62,63]. During hepatocyte repair, lipotoxicity, oxidative stress, inflammation, fibrosis, and cirrhosis pathways cause gene duplication and mutation. This mechanism may modify tumor suppressor genes like p53 and oncogenes like c-Myc, making hepatocytes tumorigenic and promoting HCC [50,64]. HCC cells are larger and more irregular than normal hepatocytes, and can be round, oval, or spindle-shaped. The cells have larger nuclei, different nucleoplasmic ratios, thicker and more irregular nuclear chromatin, and more nucleoli. Because of the increased glycogen content, HCC cells’ cytoplasm appears eosinophilic and red when stained. HCC cells grow in nested, trabecular, or vesicular patterns, as opposed to traditional hepatocyte cords. The most common type is nesting growth, in which cancer cells form nests of various sizes and many blood sinusoids provide resources for tumor growth [65,66]. HCC can produce liver enlargement due to mass development or hepatic failure in advanced stages, especially when paired with severe cirrhosis and liver atrophy. Gross symptoms differ according to the type of HCC. Nodular HCC features one or more nodules, while massive HCC has a huge bulk [66].

## 5. Recent Advancements in Targeted Therapies for MASLD

Typically, lifestyle changes, including dietary adjustments like calorie restriction, weight loss, and the reduction in hepatic fat buildup, are essential to the early prevention and long-term management of MASLD [67,68]. Hepatic lipid metabolism improves with a reduced saturated fat and trans-fat diet, especially fried meals and animal fats, and moderate unsaturated fat intake such olive oil and fish oil [52,69,70,71,72,73,74,75]. Moreover, it is advisable to regulate sugar consumption and to steer clear of high-sugar beverages and confections [76,77]. A daily exercise regimen has been shown to benefit lipid metabolism [9]. Moderate alcohol consumption is also crucial.

Additionally, pharmacological treatment is also necessary for MASLD symptom reduction and progression control, especially in the mid to late stages when lifestyle modifications are insufficient [18,77,78,79]. Targeted drugs can decrease natural cell damage, side effects, and treatment efficacy while remaining safer and more effective. Thus, small-molecule-targeted therapeutics are a hot issue in liver disease medication research and development [80]. These therapies control MASLD molecular development by targeting aberrant signaling pathways that become overexpressed or underexpressed as the disease progresses, as well as using gene-related therapies (e.g., siRNAs) to influence specific translation products.

The progression of MASLD severity arises from the interaction and continuous impact of various small-molecule targets at distinct phases.

During the initial phase of MASLD, a group of essential enzymes involved in lipid metabolism collaborate to facilitate the aberrant accumulation of triglycerides in the liver. ATP citrate lyase (ACLY), an upstream important enzyme in lipid synthesis, cleaves citric acid into acetyl coenzyme A and oxaloacetate [28,81], while Acetyl-CoA carboxylase (ACC) employs the acetyl coenzyme A provided by ACLY to catalyze the formation of malonyl coenzyme A [82]. Fatty acid synthase (FAS) utilizes malonyl coenzyme A as a substrate to engage in a series of intricate biochemical processes for the synthesis of fatty acids [25,83]. Subsequently, stearoyl coenzyme A desaturase 1 (SCD1) further converts the saturated fatty acids produced by FAS into monounsaturated fatty acids [84], which serve as crucial intermediates in the TG synthesis catalyzed by diacylglycerol acyltransferase 2 (DGAT2), finally resulting in a significant TG buildup in the liver and inducing steatosis [85]. Moreover, HMG-CoA reductase functions as a pivotal rate-limiting enzyme in the cholesterol biosynthesis pathway, facilitating cholesterol production via a sequence of events, augmenting the total intracellular lipid content, and, consequently, intensifying the hepatic fat burden [28,86]. And abnormal thyroid hormone receptor-β (THR-β) function unique to the liver causes lipid metabolism problems and encourages the buildup of fat in the liver [87,88]. The sodium-glucose cotransporter 2 (SGLT2)-mediated hyperglycemic condition enhances glucose metabolism, facilitating lipid synthesis and promoting the accumulation of fatty acids and triglycerides, hence aggravating fat deposition in the liver [89]. Simultaneously, aberrant glucagon-like peptide-1 (GLP-1) production does not adequately stimulate insulin release in a glucose-concentration-dependent manner, resulting in heightened insulin resistance (IR) [90,91]. IR diminishes the cellular sensitivity to insulin, further disrupting glucose and fat metabolism. Furthermore, peroxisome proliferator-activated receptor (PPAR) governs the oxidative metabolism of fatty acids and sustains intracellular lipid homeostasis under standard settings. In the steatosis stage, the PPAR activity was inhibited, leading to a reduction in fatty acid oxidation, obstructing the catabolic pathway of lipids [92,93,94].

As steatosis progressed, the liver advanced to the MASH stage. At this juncture, excessive fat buildup in the liver serves as a catalyst for inflammation and oxidative stress. The substantial accumulation of adipose tissue prompts immune cells to secrete tumor necrosis factor (TNF)-α, interleukin (IL), and other inflammatory mediators, which swiftly activate the inflammatory signaling pathway, inciting an inflammatory response in the liver [95,96,97]. Simultaneously, the intracellular redox equilibrium is perturbed, leading to a marked rise in the production of reactive oxygen species (ROS), which possess potent oxidative properties capable of targeting biomolecules such as proteins, lipids, and DNA, thereby causing damage to the cellular structure and function. This oxidative stress not only enhances the inflammatory signaling route but also induces endoplasmic reticulum stress, thereby activating the associated apoptotic signaling pathway, leading hepatocytes to undergo apoptosis [40,41,97]. Apoptosis signal-regulating kinase 1 (ASK1) is pivotal in this process, activated by various stress signals, including ROS, which subsequently triggers downstream apoptosis and inflammation-related signaling pathways, intensifying apoptosis and inflammatory responses, thereby creating a detrimental cycle that exacerbates liver injury [83,98,99,100]. Additionally, the Farnesoid X receptor (FXR) is a significant regulator of bile acid metabolism, and its diminished activity contributes to increased inflammation [92,101]. And the dysregulated production of fibroblast growth factors (FGFs) obstructs glycolipid metabolism and stimulates the release of inflammatory mediators, hence affecting hepatocyte function and aggravating hepatic inflammation progression [102,103,104]. Due to the ongoing inflammation and cellular damage in MASH, the liver progressively develops hepatic fibrosis. At this stage, the targets and cells related to liver fibrosis are significantly stimulated. For instance, chemokine receptor (CCR) family members are crucial in identifying and binding to specific chemokines, thereby drawing inflammatory cells and fibroblasts to the injured liver area, establishing a cellular foundation for the ensuing fibrotic process [105]. The transforming growth factor-β (TGF-β) signaling pathway is pivotal in liver fibrosis. Irregularities in these sites stimulate HSCs, inducing a phenotypic shift from dormant stellate cells to myofibroblasts characterized by elevated proliferative and secretory abilities [106,107]. These myofibroblasts secrete substantial quantities of extracellular matrix, including collagen and fibronectin. The ongoing excessive accumulation of the extracellular matrix (ECM) progressively disrupts the natural tissue architecture of the liver, resulting in diminished elasticity and compliance [108,109]. If not adequately managed, liver fibrosis will progress to cirrhosis.

Prolonged inflammation, oxidative stress, and persistent changes in the hepatic microenvironment significantly jeopardize the genetic stability of hepatocytes [61]. Due to the influence of many oncogenic stimuli, critical oncogenes, including p53, undergo mutations and become inactivated, resulting in the loss of their normal regulatory functions in cell proliferation and apoptosis. Simultaneously, proto-oncogenes are aberrantly activated, resulting in exaggerated cellular proliferation signals, disrupting the equilibrium between cell proliferation and apoptosis, leading to abnormal cellular proliferation. Concurrently, tumor cells release various cytokines and chemokines throughout their growth, which can attract immune cells, vascular endothelial cells, and others, thus establishing a unique microenvironment that facilitates tumor growth, invasion, and metastasis, namely, the HCC tumor microenvironment. Furthermore, tumor cells possess immune evasion strategies, including the expression of immune checkpoint molecules like programmed death ligand 1 (PD-L1), which interact with PD-1 on T cells, thereby inhibiting T-cell activation. The interplay of these variables ultimately results in the emergence of MASH-related HCC [50,64].

Consequently, irregularities in these small-molecule targets encompass adipose accumulation, inflammation, cellular injury, hepatic fibrosis, and cirrhosis, as well as MASH-associated carcinomas, leading to a continuum of worsening in MASLD from present to absent and from mild to severe. Consequently, the subsequent section will utilize this genealogical path as the foundational concept to summarize representative medications with confirmed efficacy against pertinent small-molecule targets in recent clinical trials, together with select preclinical research featuring innovative therapeutic concepts (Figure 2). This will offer novel alternatives for MASLD treatment and fresh insights for focused medication development.

### 5.1. Targets and Drugs for Anti-Lipid Buildup

Steatosis results from a confluence of elements, including the activation of numerous lipogenesis-enhancing enzymes or pathways in the liver under pathological conditions and IR in glucolipid metabolism [21,86,110,111,112,113,114]. This section summarizes the targeted pharmaceuticals discovered to be targeting targets linked to aberrant lipid metabolism and the resultant suppression of fat buildup. The targeted drugs for the respective targets are summarized in Table 1, following the description in this section.

#### 5.1.1. Targeted Lipid-Metabolizing Enzyme and Pathway Drugs

The major development of fatty liver lesions in MASLD arises from dysfunctions in the enzymes and signaling pathways related to lipid metabolism; hence, addressing these mechanisms with small-molecule medicines presents a possible therapeutic strategy for MASLD.

(1)ACC

ACC inhibitors selectively inhibit ACC activity and decrease TG synthesis in the liver, leading to a reduction in fatty acid synthesis and offering a therapeutic benefit for MASLD [151].

PF-05221304 functions as an inhibitor of ACC1 and ACC2. Two parallel Phase 2a clinical studies were conducted to examine the effects of ACC1/2 inhibition in the liver, involving a study population of adults with fatty liver disease. The primary clinical study indicated that PF-05221304 produced a dose-dependent decrease in fat content among patients with fatty liver disease [115]. The second study examined the therapeutic effect of PF-05221304 in conjunction with the DGAT2 inhibitor PF-06865571 after six weeks of treatment. The findings indicated that the hypertriglyceridemic effect induced by the ACC inhibitor was reduced, implying that the combination of PF-05221304 and PF-06865571 may overcome certain limitations associated with the use of ACC inhibitors alone [115]. MK-4074 is a small-molecule inhibitor that specifically targets hepatic ACC1 and ACC2. In preclinical animal models and clinical trials, the treatment of MK-4074 suppressed de novo lipogenesis (DNL) and augmented hepatic fatty acid oxidation (FAO), resulting in a substantial decrease in hepatic TG levels and a modest reduction in fibrosis in preclinical tests. The Phase 1 clinical study evaluated alterations in hepatic lipid content in adult patients with fatty liver disease following multiple oral doses of MK-4074, yielding encouraging results. The study indicated that the administration of MK-4074 for one month led to a reduction in hepatic lipogenesis in patients with hepatic steatosis, resulting in a 36% decrease in hepatic TG levels. The study observed a 200% increase in plasma TG levels, likely due to sterol regulatory element binding protein (SREBP)-1c activation and enhanced very-low-density lipoprotein (VLDL) secretion. Furthermore, both NDI-010976 [117] and GS-0976 [118], which are ACC inhibitors, have demonstrated efficacy in clinical trials in reducing steatosis in MASLD patients. However, multiple studies have shown that ACC inhibitors can predispose to hypertriglyceridemia in the treatment of MASLD, which generally limits their usage as monotherapy.

(2)FAS

The expression and activity of FAS are frequently elevated in the liver tissue of individuals with MASLD [25,83]. Inhibitors that target FAS may decrease fatty acid synthesis and improve hepatic lipid metabolism.

FT-4101, a FAS inhibitor, has been shown to inhibit hepatic de novo lipogenesis in a dose-dependent manner in two preliminary randomized clinical trials following a single administration of FT-4101. Treatment with 3 mg of FT-4101 over 12 weeks led to a significant enhancement in hepatic steatosis and a reduction in hepatic DNL. Additionally, both single and repeated doses of FT-4101 demonstrated safety and tolerability [119]. TVB-2640 (denifanstat) functions as an oral inhibitor of the FAS. A previous Phase 2 clinical trial has shown a positive effect of TVB-2640 on hepatic fat reduction in patients with steatohepatitis [120]. A multicenter, double-blind, randomized, placebo-controlled Phase 2b trial revealed that the denifanstat group showed an improvement of two or more points in the non-alcoholic fatty liver disease activity score (NAS) and a greater proportion of patients with no worsening of fibrosis compared to the placebo group (38% vs. 16%), suggesting a positive treatment response. The denifanstat group exhibited a superior remission rate for MASH, at 26% compared to 11%, and the safety assessment was relatively favorable [121].

(3)ACLY

In patients with MASLD, there is an increase in ACLY gene expression, and heightened hepatic de novo lipogenesis contributes to the progression of MASLD [28,81].

Bempedoic acid (BA, ETC1002) is an ACLY inhibitor that has shown therapeutic potential for fatty liver disease by inhibiting the hepatic PXR-SLC13A5/ACLY signaling axis, and it may serve as a supplementary therapy for patients who are intolerant to statins [152]. A Phase 3 clinical trial showed a significant decrease in low-density lipoprotein cholesterol (LDL-C) levels among fatty liver disease patients [153,154].

The most recent preclinical studies indicate that the combination of BA and the GLP-1R agonist liraglutide resulted in additional reductions in hepatic steatosis, hepatocellular ballooning, and liver fibrosis in steatohepatitis mice. This finding supports the need for further investigation of this combination therapy in the clinical management of MASLD patients [155].

(4)HMG-CoA reductase

HMG-CoA reductase is a crucial enzyme in cholesterol synthesis, facilitating the conversion of HMG-CoA to mevalonate, which represents the rate-limiting step in this metabolic pathway. Cholesterol metabolism is interconnected with fatty acid metabolism in the liver. These processes collectively influence the lipid metabolism balance and are closely associated with the development of MASLD [28,86].

Statins are inhibitors of HMG-CoA reductase, utilized to reduce cholesterol levels in blood lipids, and primarily function as lipid-lowering agents. In addition to reducing cholesterol levels, these medications may confer indirect advantages for fatty liver disease. By lowering cholesterol levels, enhancing the lipid microenvironment in the liver, and modulating fatty acid metabolism to decrease hepatic fat accumulation, they exhibit significant therapeutic potential in MASLD [156]. As demonstrated by the findings of atorvastatin, there is substantial evidence supporting the efficacy of pharmaceutical interventions in addressing hepatic steatosis [157]. An analysis of various studies indicates that statin use correlates with a decreased prevalence of steatohepatitis and fibrosis, and may potentially prevent MASLD [158]. Moreover, a meta-analysis examining the efficacy of statins in treating fatty liver disease and steatohepatitis revealed that statin use significantly reduced aspartate transaminase (AST) and alanine transaminase (ALT) levels, along with notable improvements in liver histology among patients with fatty liver disease across various clinical trials [159], particularly in those with concomitant hyperlipidemia [160]. A clinical trial also indicated that the occurrence of portal vein thrombosis (PVT) was reduced in patients with decompensated cirrhosis who were administered statins compared to those who were not [161]. Besides statins, evidence suggests that specific novel compounds may also inhibit HMGCR.

(5)SCD1

SCD1 is a pivotal enzyme in the metabolism of long-chain saturated fatty acids, such as stearic acid. SCD1 introduces a cis-double bond between the ninth and tenth carbon atoms of stearoyl-coenzyme A, thereby converting it to oleic acid. Consequently, SCD1 facilitates TG synthesis [84]. Analytical studies indicate that SCD1 expression is increased in the liver of both MASLD patients and model mice.

Aramchol is a novel fatty acid–bile acid coupling compound (FABAC) that inhibits SCD1, resulting in a reduction in liver fat. A 52-week, double-blind, placebo-controlled Phase 2b trial was conducted to evaluate the efficacy of Aramchol. The trial results indicated that a higher proportion of patients in the Aramchol 600 mg group experienced a reduction in liver fat and achieved the remission of steatohepatitis without exacerbating fibrosis (16.7% compared to 5% in the placebo group). Safety assessments confirmed that Aramchol was safe and well-tolerated [125]. A Phase III randomized controlled trial is currently being conducted to evaluate the effects of alterations in liver fat content and long-term outcomes in patients receiving Aramchol treatment.

(6)DGAT2

The DGAT2 gene is primarily expressed in the liver and acts as a catalyst in the reaction between diglycerides (DAG) and fatty acid acyl coenzyme A, leading to the synthesis of TG [85]. The expression and activity of DGAT2 are usually enhanced in MASLD patients or in animal models, which causes the liver to produce more TG. DGAT2 inhibitors have been shown to have a part in certain preclinical studies [85].

The research team developed an antisense oligonucleotide inhibitor, IONIS-DGAT2Rx, targeting DGAT2 to facilitate the enzyme-mediated degradation of DGAT2 mRNA, thus inhibiting the synthesis of DGAT2 proteins and potentially decreasing TG production. In a blinded, controlled clinical trial of IONIS-DGAT2Rx, the results indicated that patients receiving IONIS-DGAT2Rx experienced significantly greater mean reductions in liver fat content compared to the placebo group. Additionally, there were no significant changes in glycemic control or markers of insulin resistance associated with IONIS-DGAT2Rx treatment, suggesting that IONIS-DGAT2Rx did not negatively impact glycemic control, irrespective of whether diabetes mellitus was the underlying cause or a consequence of fatty liver disease. The results of the relevant index parameters indicated favorable safety outcomes [127]. IONIS-DGAT2Rx reduces liver fat like the prior ACC inhibitor First Coastal Sugar (GS-0976); however, it does not elevate serum TGs or modify lipid metrics. Additionally, ligand-guided hepatic precision-targeted delivery techniques that integrate small molecules like GalNAc3 fragments with antisense oligonucleotides and use glycoprotein receptor-mediated endocytosis in liver cells may improve drug delivery to hepatocytes and efficacy. GalNAc3-conjugated IONIS-DGAT2Rx is being investigated.

(7)THR-β

THR-β regulates hepatic lipid metabolism, which is often impaired in MASLD, hence classifying MASLD as a form of ‘hepatic hypothyroidism’ [88]. Aberrant THR-β function specific to the liver induces lipid metabolic disorders and promotes hepatic fat accumulation. THR-β is a potential target for the treatment of MASLD [87,88].

Resmetirom (MGL3196) is an oral, highly selective THR-β agonist that specifically targets the liver, exhibiting no systemic effects, which are primarily mediated in the heart and bone through THR-α [162,163]. Additionally, resmetirom became the first medication to receive conditional approval for the treatment of fibrotic MASH in March 2024 and is the first small-molecule-targeted therapy for MASLD with a high recommendation stated in the MASLD EASL-EASD-EASO Clinical Practice Management Guidelines [9]. The method via which resmetirom lowers hepatic fat in people with MASLD may rely on the re-establishment of normal mitochondrial function and enhanced beta-oxidation [164]. Two Phase III controlled clinical trials of resmetirom, namely, MAESTRO-steatohepatitis and MAESTRO-fatty liver disease, have been completed. In the MAESTRO-NASH trial, 966 patients with steatohepatitis and liver fibrosis received either resmetirom or a placebo. At the conclusion of the 52-week study, the results indicated that both the 80 mg and 100 mg resmetirom groups demonstrated significant superiority over placebo regarding steatohepatitis remission (25.9% and 29.9% vs. 9.7%, *p* < 0.001) and fibrosis improvement (24.2% and 25.9% vs. 14.2%, *p* < 0.001), with no worsening of fibrosis or fatty liver disease scores. The changes in LDL cholesterol levels in the resmetirom groups were significantly greater than those in the placebo group, with reductions of −13.6% in the 80 mg group and −16.3% in the 100 mg group, compared to 0.1% in the placebo group (*p* < 0.001). The occurrence of serious adverse events was comparable across all treatment groups [164]. In the MAESTRO-NAFLD-1 study, adult patients with fatty liver disease or suspected steatohepatitis received varying doses of resmetirom or a placebo. The 52-week study results indicated that the incidence of treatment-emergent adverse events (TEAEs) was comparable across all groups, demonstrating safety and tolerability. Additionally, the resmetirom group exhibited significant reductions in LDL-C, apoB, TGs, and liver fat content [129]. Therefore, resmetirom is demonstrated to be safe and well-tolerated, thereby supporting its continued clinical development, currently underway in the MAESTRO-NASH-OUTCOMES trial [165].

(8)PDE

Phosphodiesterase (PDE) is an enzyme family that hydrolyzes the intracellular second messenger’s cyclic adenosine monophosphate (cAMP) and cyclic guanosine monophosphate (cGMP). Isoforms like PDE3 and PDE4 are significantly associated with lipid metabolism [166], and the inhibition of PDE activity by PDE inhibitors has been shown to attenuate MASLD.

A double-blind, randomized, placebo-controlled Phase 1a study indicated that the pan-PDE inhibitor ZSP1601 exhibited favorable pharmacokinetics (PK) and safety profiles [167]. Another Phase Ib/IIa multi-dose study involving 36 patients with fatty liver disease indicated that ZSP1601 effectively improved liver biochemistry, steatosis, and fibrosis, while also demonstrating good tolerability and safety [126]. Furthermore, pentoxifylline (PTX) is an additional phosphodiesterase inhibitor that regulates cyclic adenylate concentrations and suppresses TNF-α gene transcription. A Phase 2 trial demonstrated that PTX enhanced liver histological characteristics, including inflammation and fibrosis, in MASH patients [130]. A Phase 3 trial examining the effects of PTX in MASH is now in progress (NCT05284448).

#### 5.1.2. Glucose-Lowering Drugs

IR is an important pathogenic mechanism in MASLD [21,111,112,113,114]. In a state of IR, hepatic sensitivity to insulin is reduced, leading to decreased hepatic glycogen synthesis and increased lipid synthesis [86,111]. Numerous clinical trials have shown the effectiveness of hypoglycemic agents in treating MASLD [113,168].

(1)SGLT2

Transmembrane protein SGLT2 is mostly found in the kidney’s S1 and S2 proximal tubules. Its main purpose is to carry glucose and sodium ions from the renal tubule lumen to renal tubular cells. After crossing the transporter’s basolateral membrane, glucose enters the circulation, reducing urine glucose excretion [169]. SGLT2 inhibitors reduce blood glucose levels by blocking glucose reabsorption via SGLT2 in the proximal renal tubules [89]. A significant body of research indicates that the majority of SGLT2 inhibitors effectively enhance steatosis, the inflammatory response, and fibrosis in patients with MASLD and T2DM [89,169,170,171].

Empagliflozin, an SGLT2 inhibitor, has been validated through numerous experiments. More significantly, empagliflozin has also been evaluated in several clinical studies. A controlled clinical trial demonstrated that empagliflozin substantially lowered pathological changes in the disease compared to the control group. Notable improvements were observed in steatosis (67% vs 26%, *p* = 0.025), steatosis (78% vs 34%, *p* = 0.024), and fibrosis (44% vs 6%, *p* = 0.008), along with enhancements in total cholesterol and gamma-glutamyl transferase levels [172]. In the most recent randomized, double-blind, placebo-controlled clinical trial examining the effects of empagliflozin on liver fat in patients with metabolic-dysfunction-associated steatohepatitis without diabetes mellitus, empagliflozin demonstrated a greater median reduction in MRI-Proton Density Fat Fraction (MRI-PDFF) (−2.49% vs. −1.43%; *p* = 0.025) and more significant reductions in hepatic steatosis, ALT, body weight, and ferritin, indicating the clinical potential of empagliflozin in non-diabetic metabolic-dysfunction-associated liver disease [131]. Furthermore, empagliflozin demonstrated efficacy in improving hepatic steatosis and hepatic fibrosis in a comparative efficacy trial [173] and was found to reduce intrahepatic lipid (IHL) more significantly [174]. The other is dapagliflozin. Dapagliflozin is the first approved SGLT2 inhibitor. It is effective in lowering blood glucose and can effectively lower glycated hemoglobin A1c (HbA1c). Several clinical trials have demonstrated the effective use of dapagliflozin in MASLD. A recent controlled clinical trial involving 84 patients with T2DM and fatty liver disease demonstrated that participants assigned to the dapagliflozin group exhibited a significant reduction in liver fat content (LFC) (*p* < 0.001) after 24 weeks. Additionally, improvements were observed in serum ALT, TNF-α, and IL-6 levels in this group [132]. And a meta-analysis confirmed that dapagliflozin enhances liver function and glycemic control in patients with T2DM and fatty liver disease, as indicated by reductions in ALT, AST, fasting glucose, and HbA1c [175]. Other SGLT2 inhibitors, like canagliflozin [133,134,176] and ipragliflozin [135,177,178], have been shown in clinical trials to positively influence hepatic lipid metabolism, lower ALT and AST levels, and reduce blood glucose.

(2)PPAR

PPAR is a family of nuclear receptor proteins classed as ligand-activated transcription factors, usually divided into three isoforms: α, β/δ, and γ. In the clinical progression of MASLD, the PPAR-α activity may diminish, resulting in diminished fatty acid oxidation and hepatic fat buildup, but PPAR-γ’s function in the liver is more intricate and potentially linked to hepatic steatosis and inflammatory responses. Therefore, it is anticipated that controlling the PPAR activity will enhance lipid metabolism, reduce liver inflammation, and treat various liver disorders [168,179,180,181].

Pemafibrate (K-877) is an innovative selective PPAR-α modulator that reduces TG, enhances high-density lipoprotein cholesterol (HDL-C), hence ameliorating dyslipidemia, and has certain anti-inflammatory properties. A double-blind, randomized, controlled trial demonstrated that patients with T2DM or hypertriglyceridemia treated with pemafibrate achieved significant reductions in TG, VLDL, cholesterol, and apo C-III levels, with reductions of –26.2%, –25.8%, –25.6%, and –27.6%, respectively. The treatment was associated with good safety and tolerability [137,182]. Another multicenter, double-blind, placebo-controlled, randomized Phase II research (NCT03350165) showed that pemafibrate medication for 72 weeks significantly decreased liver stiffness, LDL-C, and ALT while maintaining a satisfactory safety profile [136]. Pemafibrate was also shown in a clinical investigation to be effective in lowering liver biochemical indicators in people who drink too much alcohol [183]. Additionally, a plethora of additional clinical trials [184], trial analyses [185,186], and multicenter studies [187] have corroborated the favorable clinical efficacy of pemafibrate as a potential pharmaceutical agent for the treatment of patients with MASLD. Pioglitazone is a PPAR-γ sensitizer. A clinical trial demonstrated that patients with T2DM experienced significant improvements in hepatic steatosis, inflammatory response, and insulin resistance following treatment with pioglitazone [188]. A Phase 4 clinical trial (NCT00994682) revealed that pioglitazone significantly improved insulin sensitivity (*p* < 0.001) and lipid metabolism in the adipose tissue of diabetic patients [138]. Additionally, a review of multiple clinical trials systematically assessed the efficacy of pioglitazone in treating patients with pre-diabetes or T2DM alongside fatty liver disease [CRD42020212025]. The findings indicated that pioglitazone significantly enhanced hepatic steatosis, inflammation, and ballooning, in addition to improving insulin sensitivity and hepatic biological markers, including plasma AST and ALT levels [189]. Nevertheless, other research indicates that pioglitazone might have adverse effects linked to PPAR-γ. The team used PXL065, a novel molecule with a similar efficacy profile to pioglitazone but fewer PPAR-γ-driven side effects, in a Phase II clinical trial to overcome the drug’s side effects. The results showed that PXL065 improved patients’ hepatic fibrosis and liver fat profiles and had a generally good safety profile [190]. And MSDC-0602K is a second-generation thiazolidinedione that selectively inhibits mitochondrial pyruvate transporters while minimizing direct interaction with PPAR-γ, hence decreasing the likelihood of adverse consequences. MSDC-0602K decreased fasting glucose, insulin levels, and indicators of liver damage in a Phase 2b clinical trial [139], and the efficacy findings of a Phase 3 trial (NCT03970031) are forthcoming.

Dual/Pan PPAR agonists are more significant in disease treatment compared to PPAR-α or PPAR-γ agonists individually [168]. Lanifibranor is a pan-PAR agonist that influences critical metabolic, inflammatory, and fibrogenic processes in the development of MASLD. A six-month Phase IIb study involved 247 patients with steatohepatitis who received 800 or 1200 mg/day of the drug. The results indicated that a markedly greater percentage of patients administered 1200 mg of lanifibranor achieved a minimum two-point decrease in SAF-A scores without exacerbating fibrosis compared to the placebo group (49% vs 22%). Additionally, the lanifibranor cohort exhibited reductions in liver enzyme levels and enhancements in the majority of lipid, inflammation, and fibrosis biomarkers [140,191]. Regarding safety, lanifibranor had a higher propensity for common minor side effects, including diarrhea and nausea; nonetheless, with fewer than 5% of patients discontinuing therapy due to these events, the safety profile was comparatively favorable. These results advocate for the additional assessment of lanifibranor’s efficacy in a Phase 3 clinical trial (NCT04849728) aimed at evaluating symptom alleviation and fibrosis enhancement in steatohepatitis patients over a 72-week period. Additionally, saroglitazar (dual PPAR-α/γ agonist) [141,142,192] and elafibranor (dual PPAR-α/δ agonist) [143,144,193] have demonstrated positive therapeutic effects in their respective clinical studies by reducing serum lipids and attenuating inflammatory responses. Other than this, structurally stable novel PPAR-α/δ agonists are under investigation, such as new triazolone derivatives [194]. However, it is also critical to remain aware that PPAR agonists frequently cause adverse effects such nausea, diarrhea, peripheral oedema, anemia, and weight gain, which should be taken into account throughout clinical trials [195,196].

(3)GLP-1

GLP-1 is an intestinal hormone secreted by L-cells in the small intestine in response to feeding. It plays a role in stimulating insulin secretion, inhibiting glucagon secretion to maintain blood glucose levels, and regulating lipid metabolism [90]. GLP-1 levels were diminished in patients with MASLD, thereby confirming the relationship between insulin resistance and the onset of MASLD [91]. Thus, GLP-1 receptor agonists (GLP-1RAs) are frequently utilized in diabetes management, with several studies indicating notable decreases in liver fat following treatment [90,91,197,198,199,200,201].

A daily injectable agonist of the GLP-1 receptor is dulaglutide. Dulaglutide dramatically decreased liver fat content (LFC) and raised GGT levels in fatty liver disease patients in a clinical trial. Liver stiffness, AST, and ALT levels also decreased moderately [149]. Semaglutide, a GLP-1RA, was assessed for its efficacy and safety in a 72-week, double-blind, Phase 2 trial involving patients with steatohepatitis and varying levels of liver fibrosis. The results indicated that the semaglutide group demonstrated significant superiority over the placebo group regarding the primary endpoint, which was the remission of steatohepatitis without fibrosis worsening (40%, 36%, and 59% compared to 17% for the 0.1, 0.2, and 0.4 mg groups, respectively) [145,202]. However, in terms of safety, the incidence of nausea, constipation, and vomiting was greater in the high-dose group (0.4 mg) compared to the placebo group, with malignancy occurring in 1% of patients. Further clinical trials are necessary in order to establish clinical efficacy and safety. A Phase III clinical trial of semaglutide is currently in progress (ESSENCE, NCT04822181), with recruitment for the MASLD baseline population recently completed [203]. Efinopegdutide is a dual GLP-1/glucagon (GCG) receptor agonist, promoting insulin secretion from islet β-cells via the activation of the intracellular cAMP-PKA and PI3K-Akt pathways. Efinopegdutide’s extra agonism at GCG receptors, which increases lipolysis in adipose tissue and improves fatty acid oxidation in the liver and muscle, is more effective than GLP-1 agonists alone at promoting energy expenditure [146]. In a Phase IIa randomized, efficacy-comparative, controlled trial (NCT04944992), participants were assigned to either an efinopegdutide group (10 mg/week) or a semaglutide group (1 mg/week). At the conclusion of the trial at week 24, the primary endpoint, LFC, demonstrated a reduction. The efinopegdutide group demonstrated a significantly greater reduction compared to the semaglutide group (72.7% vs 42.3% reduction rate, *p* < 0.001). Additionally, the efinopegdutide group exhibited a marginally superior effect on body weight (8.5% vs 7.1%; *p* = 0.085) [146]. However, efinopegdutide exhibited a marginally increased incidence of adverse events and drug-related adverse events, primarily associated with an imbalance in gastrointestinal adverse events, necessitating further observation in subsequent clinical trials. Other agents, including Survodutide (a dual GLP-1/GCG receptor agonist) [204], Cotadutide (a dual GLP-1/GCG receptor agonist) [148], and Tirzepatide (a dual GLP-1R/glucose-dependent insulinotropic polypeptide receptor agonist) [205], have demonstrated encouraging outcomes in clinical trials. Besides GLP-1RA, GLP-1 analogues can fulfill similar physiological roles. Liraglutide is a GLP-1 analogue utilized as an antidiabetic medication by stimulating insulin production. A preliminary double-blind, randomized, placebo-controlled Phase 2 study demonstrated that a higher proportion of steatohepatitis patients receiving liraglutide attained remission and regression objectives compared to the placebo group (39% vs 9%, *p* = 0.019); additionally, the liraglutide cohort exhibited a reduced incidence of fibrosis progression (9% vs 36%, *p* = 0.04) [147,206]. Adverse events were mild and infrequent and relatively well-tolerated. And clinical trials have shown that the combination of liraglutide and metformin enhances body weight, intrahepatic lipid (IHL), and visceral adipose tissue (VAT), alongside glycemic control in patients [207,208]. However, several trials have revealed that liraglutide is prone to triggering gastrointestinal problems and other adverse events [208], so liraglutide needs to be studied further in depth and over a long period of time.

Furthermore, certain clinical trials indicate that combining various hypoglycemic agents may provide greater benefits compared to the use of individual agents in the treatment of MASLD. In a clinical trial evaluating the combination of tofogliflozin and pioglitazone for treating hepatic steatosis in patients with fatty liver disease and T2DM, the combination therapy demonstrated superior improvements in steatosis, liver stiffness, alanine aminotransferase levels, and glycated hemoglobin compared to either monotherapy [209]. A clinical trial (DURATION-8) proved that a once-weekly combination therapy with exenatide and dapagliflozin was superior to monotherapy in enhancing indicators of hepatic steatosis and fibrosis in individuals with T2DM [210]. And the comparison of empagliflozin and pioglitazone, both in combination with metformin (MET), for treating patients with T2DM and fatty liver disease revealed a significant reduction in lipid levels and liver enzyme tests when compared to the placebo (*p* < 0.05). Furthermore, both treatments significantly reversed the fibrotic phase of fatty liver disease (*p* < 0.05) [211], but only patients in the EMPA group had significant reductions in weight and BMI. In addition, a 52-week, double-blind, randomized, parallel, active-controlled study (ChiCTR2300070674) evaluating the efficacy and safety of semaglutide in combination with empagliflozin versus monotherapy in patients with fatty liver disease and T2DM is ongoing [212]. Additionally, by covalently attaching tesaglitazar, a PPARα/γ dual agonist, to a GLP-1RA, a research team created GLP-1RA/tesaglitazar. Preclinical investigations showed that this combination enhanced glycemic control more effectively than either GLP-1RA or tesaglitazar alone [213]. Thus, a possible future approach to treating MASLD may involve the use of multiple hypoglycemic medications or hypoglycemic medications combined with other lipid-lowering analogues.

### 5.2. Drugs and Targets for Inflammation and Cell Damage

The advancement of MASLD leads to an excessive buildup of hepatic fat, which incites stress responses including inflammation, oxidative stress, and cellular damage [40,41]. The inflammatory responses stem from the interaction of many inflammatory factors and pathways, with the inflammatory response and hepatocyte injury being interconnected, thus creating a detrimental cycle that propels MASLD towards MASH [96,97]. Consequently, targeting specific small chemical entities within the inflammatory pathways represents a viable approach for addressing the inflammatory phase of MASLD. This section outlines the targeting of inflammatory-response-related targets and the corresponding pharmacological agents in recent years. Table 2 is appended following this section and contains the targeted drugs for the corresponding targets.

(1)CCR

CCRs are G-protein-coupled receptors (GPCRs) needed for immune cell motility, inflammatory reactions, and other physiological and pathological processes. The small-molecule cytokine chemokines stimulate the directed chemotactic motions of nearby responsive cells, whereas their receptors are crucial for their action [105]. Therefore, using CCR antagonists to reduce inflammatory leukocyte infiltration may help control MASH [228].

Cenicriviroc (CVC) is a bifunctional CCR antagonist that can block both CCR2 and CCR5 [229]. CCR2 is predominantly expressed on the surface of monocytes and macrophages, and CVC can impede the chemotaxis of these cells to the site of liver inflammation while also inhibiting their production of several inflammatory cytokines that initiate the inflammatory response by obstructing CCR2. CVC may impede the chemotaxis of monocytes and macrophages to the liver inflammation site by obstructing CCR2, hence limiting the release of substantial quantities of inflammatory cytokines and initiating inflammatory reactions [230]. Preclinical studies indicated that CVC significantly reduced fibrosis in steatohepatitis in a rat model of liver fibrosis [231]. In an additional Phase 2b clinical trial (NCT02217475), the findings indicated that CVC enhanced fibrosis and demonstrated superior efficacy in advanced fibrosis [232]. Furthermore, CVC therapy exhibits anti-fibrotic and anti-inflammatory effects by decreasing various systemic inflammatory biomarkers, such as C-reactive protein, IL-6, IL-1β, and fibrinogen. A recent study indicated that CVC is safe and well-tolerated in patients with steatohepatitis and stage 0–4 liver fibrosis, thereby reinforcing its application in the clinical management of MASH [233].

(2)FXR

FXR modulates bile acid, lipid, and glucose homeostasis target genes during feeding to regulate metabolism and inflammation [234]. Bile acid metabolism disruption by FXR functional depression causes hepatic lipid buildup, inflammation, and fibrosis [92,101]. Consequently, FXR agonists are promising MASLD treatments [235].

Obeticholic acid (OCA) is a selective FXR agonist formed by adding an ethyl group to chenodeoxycholic acid, a naturally occurring FXR agonist in humans. The following positive impacts have been observed: metabolic control, suppressing inflammation, and inhibiting fibrogenesis [236]. In a multicenter, randomized, double-blind, placebo-controlled Phase III trial of OCA (REGENERATE trial), an 18-month interim analysis demonstrated that patients with steatohepatitis and fibrosis treated with OCA (10 and 25 mg/day) exhibited superior fibrosis improvement compared to the placebo group. The results indicated that the OCA groups outperformed the placebo group, with an 18% improvement in the 10 mg OCA group and a 23% improvement in the 25 mg OCA group compared to a 12% improvement in the placebo group. Consequently, OCA 25 mg is deemed to significantly enhance fibrosis and a critical aspect of steatohepatitis disease activity in patients with steatohepatitis [215,237]. In terms of adverse events, OCA was primarily associated with dose-dependent pruritus and elevated LDL-C, so a comprehensive efficacy and safety analysis at the end of the 48-week long-term trial is required before making a final assessment of its clinical efficacy. Moreover, alternative FXR agonists have demonstrated positive outcomes in clinical trials. Tropifexor (TXR) [216,238], cilofexor (GS-9674) [217,239], EDP-305 [240], norursodeoxycholic acid [241], and HTD1801 (BUDCA, or berberine ursodeoxycholate) [218] have demonstrated favorable outcomes in various clinical trials, including reductions in ALT and AST levels, diminished inflammatory responses, lowered serum lipids, and enhanced blood glucose levels. However, FXR agonists are associated with side effects, including pruritus [242], necessitating the ongoing assessment of safety and tolerability in clinical trials. Consequently, the investigation of structurally optimized synthetic FXR agonists is ongoing. MET642 is an agent that has demonstrated a lower incidence of side effects, including pruritus, in certain trials when compared to OCA. It is currently undergoing a Phase 2 clinical trial (NCT0477396) to further validate its efficacy.

(3)FGFs

FGFs are a category of cell growth factors that exhibit several biological roles, including the reduction in adipogenesis and regulation of bile acid metabolism, and possess anti-inflammatory and antifibrotic properties [104]. Consequently, targeting agonizing FGFs is a potential therapeutic approach for MASLD [243].

Pegbelfermin (BMS-986036) is an FGF21 agonist. The FALCON 1 and 2 studies were Phase 2b, multicenter, double-blind, placebo-controlled, randomized trials of pegbelfermin in patients with steatohepatitis with stage 3 liver fibrosis (FALCON 1) or compensated cirrhosis (FALCON 2), and showed that pegbelfermin improved both hepatic steatosis and fibrosis in steatohepatitis patients [219,220]. Efruxifermin, a long-acting Fc-FGF21 fusion protein that mimics the agonism of FGF21 on fibroblast growth factor receptor 1c, 2c, or 3c, is a novel targeted therapeutic designed to mimic the biological activity of FGF21 [244,245]. In a Phase 2a clinical trial, efruxifermin effectively reduced the hepatic fat fraction (HFF) in individuals with steatohepatitis (stages F1–F3) with a tolerable safety profile [221,245]. A recent multicenter, randomized, double-blind, placebo-controlled Phase 2b study (HARMONY) involving steatohepatitis patients with moderate (F2) or severe (F3) fibrosis demonstrated that the efruxifermin group exhibited superior outcomes regarding at least one stage of fibrosis improvement without a worsening of steatohepatitis at week 24, and the treatment was well-tolerated [246], supporting further evaluation in a Phase III trial. Another updated Phase II study assessed the safety and efficacy of efruxifermin in conjunction with GLP-1RA in patients with MASH, fibrosis, and T2DM. The findings indicated that the combination therapy led to a more pronounced reduction in HFF compared to GLP-1RA monotherapy. There was a significant difference (65% vs 10%, *p* < 0.001) observed, along with improvements in glucose and lipid profiles, liver injury, and markers of liver fibrosis. And it was safe and well-tolerated [247]. Analogues of FGF19, such as Aldafermin, have demonstrated efficacy in clinical trials by reducing hepatic lipid accumulation, ALT, AST, and other parameters [222,223,248].

(4)ROS

The excessive accumulation of lipids leads to the overproduction of ROS from different sources, which has a promoting effect in the development of MASLD [25,26,249,250]. Consequently, the administration of some antioxidative stress medications can diminish hepatic cellular injury and alleviate the progression of MASH.

Vitamin E is a fat-soluble antioxidant comprised of two primary groups: tocopherols and tocotrienols. In MASH, vitamin E mitigates oxidative stress by supplying hydrogen atoms that reduce and scavenge free radicals, thereby preventing lipid peroxidation induced by these radicals, which can result in liver cell damage. Vitamin E regulates intracellular signaling pathways by inhibiting the activation of specific pro-inflammatory pathways and decreasing the production of inflammatory cytokines, such as TNF-α and IL-6, thereby reducing liver inflammation [251,252,253]. In a prior large, randomized, double-blind, controlled Phase 3 clinical trial, the vitamin E group demonstrated a significantly greater enhancement in steatohepatitis scores compared to the placebo group (43% vs 19%). Additionally, the combination of pioglitazone and vitamin E resulted in improved histological hepatic steatosis and hepatic lobular inflammation [224]. In a subsequent double-blind, active-controlled trial, patients with fatty liver disease treated with delta-tocotrienol exhibited enhancements in markers of liver steatosis, oxidative stress, and insulin resistance, with delta-tocotrienol proving more efficacious in mitigating stem cell inflammation and apoptosis [254]. And, in the pooled analysis, vitamin E was recognized as the most likely intervention for attaining steatohepatitis remission and showed a significant advantage over the placebo in enhancing stage ≥1 fibrosis [255]. Furthermore, vitamin E shown greater efficacy in diminishing AST and ALT indicators [256].

Alpha-lipoic acid (ALA) is an antioxidant that occurs naturally. The accumulation of lipids and the subsequent development of inflammation in the liver have been found to be inhibited by it [257,258]. The results of a clinical investigation demonstrate that oral ALA supplementation over 12 weeks led to enhancements in lipid buildup, insulin resistance, and blood lipocalin levels in individuals [259]. In a supplementary clinical experiment, blood lipocalin and IL-6 levels showed greater improvement in the cohort of patients who incorporated ALA compared to those receiving vitamin E alone [225]. Another clinical investigation indicated that ALA treatment significantly improved insulin sensitivity and resulted in a marked decrease in ALT and AST plasma levels among participants [260].

(5)ASK1

ASK1, a mitogen-activated protein kinase kinase kinase (MAP3K), is critical for intracellular stress signaling. Inflammatory cytokines, oxidative damage, and endoplasmic reticulum stress all activate it. When activated, ASK1 phosphorylates and stimulates the downstream JNK and p38 MAPK signaling pathways. These regulate apoptosis, inflammation, and cell proliferation [83,98,99,100]. Thus, ASK1 inhibitors can help to minimize apoptosis.

Multiple prior studies indicate that selonsertib, an ASK1 inhibitor, diminishes fibrosis in patients with steatohepatitis accompanied by fibrosis [261]. In a Phase 2 clinical trial, selonsertib demonstrated efficacy in reducing steatohepatitis and stage 2–3 liver fibrosis in patients [226]. However, subsequent Phase 3 trials investigating the efficacy of 48 weeks of selonsertib treatment in reducing fibrosis among patients with steatohepatitis and advanced hepatic scarring did not achieve their primary efficacy endpoints, indicating that selonsertib did not effectively reduce fibrosis in these populations [227]. Therefore, further trials are still needed for specific clinical effects.

### 5.3. Anti-Hepatic Fibrosis Targets and Drugs

Elevated inflammation, oxidative stress, and hepatocellular damage during the MASH phase result in the progressive activation of HSCs, ultimately leading to their transformation into myofibroblasts, which subsequently synthesize and secrete substantial quantities of ECM [108,109,262]. Consequently, HSCs and ECM represent promising targets for the management of hepatic fibrosis. This section delineates recent advancements in small-molecule therapeutics aimed at HSCs or ECM (Table 3), which have yet to undergo clinical trials but offer innovative perspectives. Furthermore, based on the stage of liver fibrosis and cirrhosis, combination therapy emerges as the primary treatment modality, which is also encapsulated in the latter part of this paragraph. The integration of these two strategies signifies the future direction in the treatment of liver fibrosis and cirrhosis.

#### 5.3.1. Anti-Activation HSC or ECM Therapy

Nanoparticle delivery systems represent a novel approach to drug administration, utilizing particles sized between 1 and 1000 nanometers to encapsulate and transport agents like siRNAs to specific targets [273]. This technology enhances drug stability against enzymatic degradation within the body and improves targeting efficacy. Consequently, nanoparticle delivery, in conjunction with HSC or ECM targets, emerges as a promising therapeutic strategy for liver fibrosis.

(1)HSP47 silencing reduces ECM production

aHSCs express heat shock protein 47 (HSP47), which promotes collagen synthesis and accelerates liver fibrosis. A clinical trial with lipid nanoparticle BMS-986263 delivering HSP47 siRNA (NCT03420768) in HCV-SVR patients showed reduced collagen, hepatic fibrosis, METAVIR, and Ishak scores, and good tolerability [263]. Another team used the first innovative synthesis of AA-T3A-C12/siHSP47 LNP, an anisamide that is highly expressed on rapidly proliferating HSCs, to deliver siRNAs to liver HSCs. The experiment tested LNP-mediated siRNA delivery to liver HSCs. In the preclinical model test, AAT3A-C12/siHSP47 LNP-treated fibrotic mice had a 65% lower HSP47 protein expression and no aberrant hepatotoxicity indicators, indicating safety [264].

(2)Inhibition of autophagy in aHSCs

Autophagy suppression in aHSC also prevents liver fibrosis. The research team used the small-molecule autophagy inhibitor hydroxychloroquine (HCQ) to deliver HCQ@Retinol Liposome Nanoparticles (HCQ@ROL-LNPs) to aHSCs without damaging other liver cells. In mice models of TGF-β-induced stellate cell activation and thioacetamide (TAA)-induced hepatic fibrosis, HCQ@ROL-LNPs significantly reduced ECM deposition with good homogeneity and stability. Moreover, targeting the unique aHSCs reduced liver cell damage. The results demonstrated that mouse liver ECM deposition was considerably decreased with good uniformity and stability [265].

(3)Co-targeting macrophages and HSCs

A research team developed Fe_3_O_4_ nanoparticle (IONP)- and ferulic acid (FA)-co-encapsulated poly (lactic-hydroxyglycolic acid) (PLGA) nanoparticles (NPs) with cRGD peptides (cRGD-PLGA/IOFA) for MRI-traceable liver imaging using integrin αvβ3, which is overexpressed on fibrotic livers. In preclinical studies, IONPs encapsulated in cRGD-PLGA/IOFA had excellent MRI performance and downregulated liver Ly6Chigh/CD86+ macrophage ratios to promote collagen degradation. FA encapsulated in the same manner was also effective in reducing oxidative stress and inhibiting HSC activation. In mice, IONPs and FA decreased hepatic fibrosis synergistically. Due to its high efficiency and low toxicity, the cRGD-PLGA/IOFA-mediated co-administration of IONPs and FA demonstrated good anti-hepatic fibrosis effects in both early and late stages, which may benefit the clinical liver fibrosis patient [266].

(4)Blocking the activation of HSCs

A research team also constructed NP-AEAA-coated siRNA (si@NP-AEAA) using nanoparticle NP-AEAA, which is predominantly target-enriched in aHSCs, to regulate HSC activation and fibrotic remodeling by blocking IL-11/ERK signaling in a mouse model of hepatic fibrosis’ increased IL-11 expression. A mouse model showed that si@NP-AEAA inhibited HSC activation and reduced fibrosis and inflammation [267].

(5)Preventing HSC-KC crosstalk

miR-155 targets signaling pathways or genes to increase hepatic stellate cell growth. Thus, anti-miR-155 can minimize liver fibrosis by inhibiting HSC proliferation and ECM-producing cell production. Crosstalk between KCs and HSCs is crucial to liver fibrosis. Targeted medicines that block KC-HSC crosstalk may be effective. The study team created modular nanosystemic polymers with CXCR4-inhibitory moieties to encapsulate anti-miR-155 and limit HSC-KC interaction by blocking CXCR4 signaling in active HSCs and decreasing miR-155 expression in KCs. Fibrosis model mice showed positive antifibrotic effects in preclinical trials [268]. Another team designed a fluoropolymer to efficiently deliver anti-miR-155 to HSCs and KCs with high sorafenib drug loading. By loading sorafenib into fluoropolymer FP2M/PBA, Sor NPs achieve improved liver targeting and efficiently deliver anti-mir-155 to KCs in vivo. In chemo-gene treatment, sorafenib and anti-miR-155 may change pro-inflammatory M1 to anti-inflammatory M2 in KCS and decrease HSC proliferation. Feedback on the experimental results is welcome [269].

(6)Collagen degradation and inhibition of HSC activation

Moreover, a research team designed a glycyrrhetinic acid (GA)-type I collagenase (Col)-modified pre-drug (COL-HA-GA, abbreviated as CHG) to improve drug targeting because HSCs overproduce ECM collagen, which hinders drug delivery and reduces therapeutic efficacy. CHG micelles effectively degrade pericellular collagen and attenuate hepatic fibrosis. It also penetrates the ECM. CHG was co-delivered with sorafenib (SORA/CHG, abbreviated as S/CHG) to treat hepatic fibrosis by stimulating ECM disintegration and HSC targeting [270]. Drug targeting in the liver for liver fibrosis is another way to target breakthroughs through the unusually dense ECM. The research team hypothesized that the kondroitin-sulphate-mediated transport of nanoparticles to hepatic stellate cells by the cell surface glycoprotein CD44 would allow them to design multilayered 50 nm nanoparticles encapsulating collagenase and silymarin (COL+SLB-MLP). In preclinical studies, MLPs internalized targeting, protected COL from early plasma inactivation, broke down collagen in the hepatic ECM, and disrupted the dense collagen matrix for easier drug delivery. Silymarin (SLB) reversed HSC activation and inhibited proliferation. In mice, the medication system synergistically inhibited hepatic fibrosis [271].

(7)Reopening of LSEC to inhibit HSC activation

And, in liver fibrosis patients, capillarized sinusoids block medication transport and accelerate HSC activation and fibrosis progression by blocking blood–Disse interstitial space exchange. The researchers pre-treated liver sinusoids with riociguat, a soluble guanylate cyclase stimulator, and peptide-nanoparticle-mediated insulin growth factor 2 receptor-mediated targeted delivery of JQ1, an antifibrotic medication. Riociguat reversed hepatic sinusoidal capillarization and maintained the relatively normal porosity of LSECs, allowing IGNP-JQ1 to pass through the endothelial wall and accumulate in Disse’s interstitial space. Thus, the activated HSCs selectively took up IGNP-JQ1, limiting their proliferation and hepatic collagen deposition. CTC-induced fibrotic mice showed substantial fibrosis reduction with this combination. Thus, restoring LSEC fenestration is a possible liver fibrosis treatment [272].

#### 5.3.2. Holistic Therapy

At the level of liver fibrosis and cirrhosis, pharmacological therapy alone is frequently insufficient, necessitating a comprehensive treatment approach. The most notable progress in the field is the emergence of small-molecule-targeted medication therapy, as previously outlined. This unique strategy is utilized solely to prolong the lives of critically ill patients, alleviate their symptoms, and improve their prognosis and quality of life through comprehensive therapeutic procedures. Fundamental treatment encompasses dietary assistance and fluid management for patients; addressing the underlying cause, such as ongoing antiviral therapy for individuals with hepatitis B cirrhosis [274]; symptomatic management, including the administration of diuretics (e.g., spironolactone and furosemide) for ascites in portal hypertension, or peritoneal drainage to alleviate fluid accumulation [275,276,277]; the management of consequences, including the elimination of the etiological factor, control of gastrointestinal hemorrhage, and alleviation of hepatic encephalopathy with agents such as lactulose [278,279,280,281]; and the treatment of hepatorenal syndrome, which entails the administration of terlipressin alongside albumin to augment renal perfusion. Hemodialysis is a treatment option that can be used when renal replacement therapy is judged to be required [282,283,284,285]; for emergencies such as rupture of esophagogastric fundic varices, growth inhibitors and their analogues can be used; and, in critical cases, subendocardial therapy or surgery is also mandatory, including esophageal variceal ligation (EVL), and transjugular intrahepatic portosystemic shunt (TIPS) [286,287,288]; and liver transplantation should be contemplated for individuals with end-stage cirrhosis who demonstrate profound hepatic dysfunction, and substantial comorbidities, and have had many unsuccessful treatment modalities [289,290,291,292].

### 5.4. Targeted Therapy for MASH-Related HCC

In contrast to general HCC, the pathogenesis of MASH-related HCC encompasses recurrent lipotoxicity, inflammation, fibrosis, and cirrhosis. Consequently, inflammation, oxidative stress, insulin resistance, and other metabolic abnormalities may serve as therapeutic targets, with previously discussed related pharmacological interventions not reiterated here. This section concentrates on pharmaceuticals that selectively engage receptors associated with HCC-related processes. This section initially summarizes the existing HCC-targeted agents with established efficacy, subsequently addressing innovative drug therapies that have demonstrated promising outcomes in preclinical studies, primarily employing novel small-molecule-targeting technologies or unconventional targets, potentially offering new insights and directions for future targeted therapies in hepatocellular carcinoma (Table 4). Ultimately, the current cancer staging for individuals with MASH-related HCC is summarized alongside the associated combo therapy.

#### 5.4.1. Small-Molecule-Targeted Drugs for HCC with Proven Efficacy

The systemic treatment of HCC has progressed from singular targeted drug therapies, such as sorafenib and lenvatinib, to combination targeted drug therapies incorporating immune checkpoint inhibitors, exemplified by atezolizumab plus bevacizumab. This evolution has led to the introduction of novel molecularly targeted monotherapies, such as donafenib, new immuno-oncology monotherapies like durvalumab, and innovative combination therapies, including tremelimumab and the durvalumab plus tremelimumab regimen, which are demonstrating promising outcomes in clinical trials [308].

Sorafenib, the first small-molecule-targeted liver cancer drug, is a major advance. Sorafenib inhibits signal transmission and tumor cell growth by targeting many intracellular and cell surface kinases, including Raf-1 and B-Raf. Sorafenib also inhibits vascular endothelial growth factor receptors (VEGFR)-2 and VEGFR-3, which reduce the tumor blood supply [294]. A lot of clinical evidence shows that sorafenib improves advanced HCC survival [294,309,310,311]. This approval revolutionized targeted liver cancer therapy, giving advanced liver cancer patients a new option. This drug is the main treatment for advanced HCC in many clinical studies and practice guidelines. Sorafenib was approved by the Food and Drug Administration (FDA) to treat unresectable HCC on November 16, 2007, marking a major advance in this important health concern. This approval revolutionized targeted liver cancer therapy, giving advanced liver cancer patients a new option. This drug is the main treatment for advanced HCC in many clinical studies and practice guidelines.

Lenvatinib is an oral multi-targeted tyrosine kinase inhibitor (TKI) that primarily exerts anti-angiogenic effects by inhibiting VEGFR 1–3, fibroblast growth factor receptors (FGFR) 1–4, and platelet-derived growth factor receptor (PDGFR)-α, as well as RET and KIT kinases, which are crucial for the proliferation, survival, and metastasis of HCC cells [312]. A substantial amount of clinical research has shown the equivalent efficacy of lenvatinib and sorafenib. A subsequent investigation has demonstrated enhanced efficacy for lenvatinib. Furthermore, research has shown lenvatinib’s substantial effect on patient survival, exhibiting a notable capacity to regulate tumor growth [313,314,315,316,317]. Based on these findings, leading clinical recommendations both nationally and internationally now endorse lenvatinib as the optimal first-line therapy for HCC. In August 2018, the U.S. FDA sanctioned lenvatinib for use as a first-line therapy for patients with advanced liver cancer.

Donafenib is a deuterated variant of sorafenib, an orally administered small-molecule anticancer agent that functions as a multi-target, multi-kinase inhibitor. It operates by obstructing the cell-surface kinases linked to VEGFR, PDGFR, FGFR, and RAS/RAF signaling pathways. Consequently, tumor cell proliferation and tumor angiogenesis are inhibited [318]. It has been demonstrated to have favorable HCC inhibitory effects in several clinical trials [319,320]. For instance, a multicenter Phase II/III clinical trial evaluating the efficacy and safety of donafenib versus sorafenib in patients with advanced HCC revealed a significantly longer median overall survival (12.1 months vs. 10.3 months), as well as comparatively higher rates of disease control, objective remission, and grade ≥3 adverse events in the donafenib cohort relative to the Sorafenib group [321]. Moreover, drug-related adverse events were significantly reduced in the donafenib group. On 8 June 2021, the Chinese National Medication Administration authorized donafenib for sale as a national class 1 novel medication, making it the country’s first domestically manufactured first-line targeted agent for HCC. Among other applications, it was authorized for the first-line treatment of unresectable HCC that has not yet undergone systemic therapy.

Many second-line drugs, including cabozantinib [322], regorafenib, and tislelizumab [323,324], have also shown excellent results in clinical trials for the treatment of HCC [314]. We shall refrain from offering a comprehensive description in this context. Kindly consult the table for precise information. And the therapeutic arsenal includes several combination medicines that have shown promising outcomes in various clinical trials. The combination of atezolizumab and bevacizumab has been recognized as the standard of care for the initial treatment of patients with advanced HCC, evidenced by its superior overall and progression-free survival outcomes compared to sorafenib, as demonstrated in the Phase III IMbrave150 clinical study [303]. A recent analysis conducted after 12 months revealed that atezolizumab plus bevacizumab demonstrated a median overall survival that was 5.8 months longer than that of Sorafenib [325]. And a solitary high-dose infusion of tremelimumab (a monoclonal antibody that utilizes an anti-cytotoxic T-lymphocyte-associated antigen 4 mechanism), combined with the administration of Durvalumab (an antibody targeting programmed cell death ligand-1), has demonstrated promising clinical efficacy and safety results, along with a significant improvement in patient survival. This therapy protocol is referred to as Single Tremelimumab Regular Interval Durvalumab (STRIDE) [326,327]. A Phase I/II clinical trial assessing tremelimumab and durvalumab as monotherapies and in combination for patients with unresectable HCC demonstrated a significant benefit/risk profile for both agents. The combination of tremelimumab 300 mg and durvalumab 1500 mg exhibited the most favorable benefit–risk profile, characterized by elevated CD8+ lymphocyte levels and the longest median overall survival [325,328]. A multitude of combination applications have been developed, including cabozantinib plus atezolizumab [322,329], sintilimab plus a bevacizumab biosimilar (IBI305) [330], lenvatinib plus pembrolizumab [315,331,332,333,334], camrelizumab plus apatinib [335], nofazinlimab plus lenvatinib [336], nivolumab plus ipilimumab [337,338,339], penpulimab plus anlotinib [340], atezolizumab plus lenvatinib [341], toripalimab plus bevacizumab [342,343], and toripalimab plus lenvatinib [344,345], all of which have achieved favorable HCC therapeutic effects in clinical trials.

#### 5.4.2. Innovative Small-Molecule-Targeted Therapies

Furthermore, numerous novel small-molecule-targeted medicines have emerged in recent years, yielding promising outcomes.

(1)Proteolysis-targeting chimeras (PROTAC)

PROTAC is a heterobifunctional molecule comprising two ligands connected by a linker; one ligand binds to the target protein while the other targets an E3 ligase. The E3 ligase facilitates the attachment of a ubiquitin molecule to a lysine residue on the target protein, resulting in polyubiquitination. Proteins adorned with polyubiquitin chains are subsequently recognized and degraded by the 26S proteasome, thereby reducing the distance between the target protein and the E3 ligase and promoting the ubiquitination of the target protein [346,347,348,349,350]. In contrast to the conventional ‘occupancy-driven’ mode, PROTAC operates in an event-driven manner, capable of binding to any site on the target protein and inducing its degradation without requiring high affinity, hence offering a highly efficient catalytic degradation mechanism [351]. And the advent of PROTAC as a novel therapeutic approach offers the potential to overcome the challenge of resistance to small-molecule medicines [346,352,353].

A research team developed the liver-targeted chimeric (LIVTAC) strategy, utilizing liver parenchymal cells that express the anti-sialylated glycoprotein receptor (ASGPR) at elevated levels. They linked PROTAC molecules to ASGPR and created XZ1606, a mammalian bromodomain and extra-terminal domain (BET)-targeted LIVTAC drug, facilitating hepatic targeting of PROTAC. The PROTAC molecule infiltrates the cell through ASGPR-mediated endocytosis and is cleaved by histone B within the lysosome, liberating the PROTAC molecule, which subsequently degrades the target protein (i.e., the protein of interest, POI). The drug molecule efficiently degrades BRD4, a target protein in the BET family, thereby facilitating apoptosis and inhibiting the growth of HCC cells. In the Huh-7 xenograft mouse model, the XZ1606-treated group demonstrated a remarkable tumor growth suppression of 58.1%, corroborated by the effective degradation of BRDs and downregulation of c-Myc in the tumor. The scientists additionally co-administered XZ1606 (1.5 mpk, q3d) and sorafenib (30 mpk, q2d), resulting in the full inhibition of tumor growth, with later experiments indicating that the combination greatly extended overall survival in mice. The safety assessment revealed that XZ1606 possesses a favorable safety profile [304]. Therefore, it is anticipated that experimental research on the application of PROTAC in the future clinical management of HCC would persist.

(2)Antibody–drug conjugate (ADC)

ADC is an innovative class of targeted therapeutic agents including a monoclonal antibody, a conjugate, and a cytotoxic medication that administers extremely precise and potent cytotoxic agents to tumors. In this method, the antibody guides the ADC to the tumor cells, where it selectively binds to the target antigen and releases the cytotoxic chemical to promote tumor cell death [354,355,356]. GPC3, which is prominently expressed in HCC, served as a target for the creation of GPC3-specific ADCs. In preclinical studies, GPC3-specific ADCs showed significant efficacy in GPC3-positive Hep3B and A431-GPC3 cells, exhibiting robust tumoricidal action and highlighting the therapeutic promise of ADCs for HCC [305].

Additionally, the dysregulation of CLDN6 additionally fosters resistance to sorafenib by causing a phenotypic transition of HCC cells towards a biliary lineage [305,357]. The research team confirmed the anti-tumor activity of this ADC both as a monotherapy and in conjunction with sorafenib by integrating an anti-CLDN6 monoclonal antibody with the cytotoxic drug DM1 (CLDN6-DM1). Several ADCs are presently in clinical assessment for safety and tolerability in the management of advanced HCC [306].

(3)A3 adenosine receptor (A3AR)

A3AR is typically overexpressed in HCC cells and underexpressed in normal hepatocytes, rendering A3AR a potential diagnostic marker and therapeutic target for tumors.

Namodenoson (CF102), a synthetic small-molecule agonist, preferentially activates A3AR, modulating the PI3K/AKT, NF-kB, and Wnt signaling pathways, resulting in the apoptosis of HCC cells, and has demonstrated the ability to decrease HCC growth in preclinical tests [358]. A Phase 1/2 trial established the safety and favorable tolerability of Namodenoson; additionally, a Phase 1/2 study involving patients with advanced HCC revealed the beneficial anticancer benefits of Namodenoson [307]. A Phase 3 trial of namodenoson in HCC is now underway (NCT05201404), owing to its demonstrated efficacy and favorable safety profile in HCC patients, with additional clinical uses anticipated.

#### 5.4.3. Comprehensive Treatment for HCC

Due to the progressive and distinctive characteristics of MASH-related HCC, integrated planning and diversified comprehensive treatment are essential throughout the treatment process, considering the patient’s current condition and the specific stage of HCC. In the early and intermediate stages, surgical resection, ablation therapies (such as radiofrequency ablation (RFA) and microwave ablation (MWA), which precisely target tumor tissue while minimizing damage to adjacent healthy tissues), and liver transplantation may be conducted if conditions allow [359]. Following this, the previously described small-molecule-targeted medicines are integrated based on criteria such as high-risk factors and malignancy level [360,361]. Transcatheter artery chemoembolization (TACE) is a feasible alternative for individuals with medium and advanced HCC [362]. This approach not only impedes the hepatic artery’s blood flow to control tumor growth but also promotes the sustained retention of chemotherapeutic agents in the lesion, thereby minimizing systemic toxicity and improving therapeutic effectiveness. Additional therapeutic modalities, including radiotherapy techniques (notably, advanced yttrium-90 radiation therapy) and local ablation procedures, are also utilized [363,364]. These modalities are subsequently combined with the previously mentioned small-molecule-targeted treatment approaches to attain systemic therapy [365]. Therefore, when developing a treatment plan, it is essential that we evaluate the patient’s liver function, tumor stage, physical condition, and additional aspects to create a tailored therapy strategy.

## 6. Additional Metabolic Variables That Affect MASLD Progression

Alongside the primary reasons of metabolic dysregulation already listed, such as lipid accumulation, insulin resistance, and the dysregulation of inflammation and oxidative stress, additional metabolic abnormalities manifest during the progression of MASLD. For example, there exists a metabolic imbalance of gut microbiota and microRNA.

### 6.1. Metabolic Environment Associated with Intestinal Flora

The gut microbiota is a huge collection of bacteria, fungi, and viruses, with bacteria being the most common. These microbes comprise a complex ecology in the gut, with species and amounts varying by person, but they perform many critical tasks that affect human health. Through the gut–liver axis, intestinal bacteria alter liver lipid metabolism. Specific bacteria produce lipopolysaccharide (LPS), which stimulates toll-like receptor 4 (TLR4) on liver cell membranes and increases fatty acid uptake and TG synthesis. Many ways exist for gut microbiota to impact bile acid metabolism and insulin sensitivity [366]. Thus, gut flora dysbiosis causes hepatic steatosis through various metabolic pathways [367,368,369,370]. Moreover, Intestinal dysbiosis compromises intestinal barrier integrity, allowing intestinal bacteria and their byproducts (e.g., LPS) to enter the bloodstream, activating hepatic immune cells, releasing many inflammatory cytokines, inducing systemic inflammatory responses, disrupting normal hepatic metabolism, promoting hepatic inflammation, and accelerating MASLD progression from simple steatosis to MASH [371]. The gut microbiota of MASLD patients is frequently less diverse [372]. Beneficial bacteria like Bifidobacterium and Lactobacillus are decreasing while harmful bacteria like Enterobacteriaceae are increasing. A diet high in fat and sugar, prolonged alcohol use, and other unhealthy lifestyle choices can cause dysbiosis, an imbalance in gut flora makeup. These variables can alter the gut ecology, promoting some bacteria and inhibiting others. Research has found that MASLD patients have less Bifidobacterium, Bacteroides, and Ruminococcus in their guts and more Parabacteroides and Prevotella [373].

Consequently, MASLD treatment methods that increase gut microbial diversity, along with others, have become important study areas. Included are the following: the utilization of probiotics and prebiotics, fecal microbiota transplantation (FMT), and associated targeted pharmaceuticals.

(1)The utilization of probiotics and prebiotics

Live probiotics can improve human health when ingested in sufficient doses. Increasing intestinal flora diversity and beneficial bacteria, decreasing fat synthesis, lessening hepatic inflammation, and increasing antioxidant enzyme activity are all part of this. Prebiotics including dietary fiber and oligosaccharides support healthy gut flora growth. Thus, probiotics and prebiotics strengthen intestinal microbiota, reducing intestinal inflammation and modifying hepatic lipid metabolism and insulin sensitivity, treating MASLD [374].

MASLD probiotic and prebiotic supplement clinical trials have yielded promising results. In a clinical investigation, individuals with fatty liver disease who took Clostridium butyricum capsules with rosuvastatin had considerably greater Lactobacillus and Bifidobacterium levels than the control group (*p* < 0.05). The patients’ rectal bacilli levels dropped, relieving bacterial dysbiosis. Inflammatory indicators including TNF-α and IL-6, as well as hepatic steatosis and fibrosis markers like TC, TG, FFA, TBIL, ALT, AST, and PP, were significantly lower than the control group (all *p* < 0.05) [375]. Moreover, a randomized, double-blind, placebo-controlled study demonstrated that probiotic supplementation significantly reduced hepatic steatosis and fibrosis in fatty liver disease patients (*p* < 0.001). Fasting blood glucose, TG, and serum inflammatory mediator levels were decreased (*p* < 0.05) [376]. In another placebo-controlled trial, steatohepatitis patients who utilized oligofructose had more Bifidobacteria in their intestines and decreased Clostridium perfringens XI and I bacteria levels (*p* < 0.05). NAS and hepatic steatosis improved considerably with supplementation (*p* = 0.016), without adverse effects [377]. Furthermore, a clinical study found that resistant starch (RS) lowers hepatic TG and improves gut microbiota [378].

Therefore, by altering the gut flora, probiotic and prebiotic supplements may be a promising therapeutic strategy for the treatment of MASLD.

(2)FMT

FMT is a therapeutic procedure that entails the transfer of fecal microbiota from a healthy donor to the gastrointestinal tract of a patient [379]. FMT has been shown to restore gut flora diversity and function in MASLD patients. FMT has been shown to worsen hepatic steatosis, inflammation, and fibrosis, making it a promising MASLD treatment. A randomized controlled clinical research found that FMT improves gut microbiota dysbiosis and reduces liver fat in fatty liver disease patients [380]. Novel studies like fecal virosome transplantation (FVT), which rebuilds the intestinal micro-ecological environment and treats certain diseases by transplanting viral populations from healthy donors’ feces into patients’ intestines, may offer a promising treatment for metabolic diseases. The entire potential of FMT in this setting needs further validation through scientific and clinical studies [381].

(3)Associated targeted pharmaceuticals

Several of the drugs listed above can improve the gut flora by modifying body metabolism, reducing MASLD. Obeticholic acid, an FXR agonist, alters bile acid metabolism and gut flora. A number of natural medicines or herbal therapies have been found as viable MASLD treatments by improving the gut–liver axis and regulating metabolism. Diosgenin reduces fat buildup, cholesterol metabolism, fibrosis, and inflammation. Diosgenin improved the gut flora, especially Clostridium difficile (LDA score 4.94), in a preclinical experiment. It also decreased serum and liver TC, TG, ALT, and AST and regulated bile acid metabolism through the hepatic and intestine FXR-SHP and FGF15 pathways [382]. (20R)-Panaxadiol, a ginseng diol, treats obesity. Preclinical experiments showed that PD increased Muribaculaceae and Lactobacillus in model mice. The antioxidants FAD and lipoic acid increased while the inflammatory metabolites prostaglandins (PG) and LPS decreased. The intestinal flora also strongly correlated with anti-inflammatory and antioxidant metabolites [383].

Moreover, Yiqi-Bushen-Tiaozhi Recipe (YBT) [384], Yindanxinnaotong formula [385], ginsenosides [386], luteolin [387], and Gynostemma pentaphyllum (GP) [388] have all been shown to have some intestinal flora-improving effects.

### 6.2. microRNAs

MiRNAs (microRNAs) are a class of endogenous small non-coding RNAs of approximately 18–25 nucleotides in length [389]. And patients with MASLD have markedly changed expression levels of many miRNAs in their peripheral blood and liver tissue. Future MASLD treatments may target these aberrantly changed miRNAs [59,390]. Research indicates that several miRNAs are linked to the progression of MASLD, for instance, miR-199a-3p [391], miR-193a-5p [392], miR-149-5p [393], miR-411-5p [394], miRNA-10b [395], miR-665-3-p [396], miR-122 [397], miR-34a [398], and miR-742-3p [399].

Through the highly competitive binding to mature miRNAs in vivo, antagomiR, a chemically modified miRNA inhibitor, suppresses miRNAs by blocking the complementary pairing between miRNAs and their target mRNAs. This inhibitor can penetrate tissue and cell membrane barriers in vivo to accumulate in target cells, and it exhibits greater stability and inhibitory activity in both in vitro and animal models. Consequently, another potential strategy for the targeted treatment of MASLD in the future may be the application of particular antagomiR therapy against aberrant miRNAs [400,401].

## 7. Conclusions and Perspective

This review analyzes the latest definition of MASLD, focusing on its two fundamental components, ‘metabolism’ and the ‘continuous development of disease spectrum’, as well as the pathological advancement of MASLD from simple steatosis to MASH, fibrosis, and MASH-related HCC, together with extrahepatic metabolic abnormalities such as IR and altered intestinal flora. The review subsequently summarizes metabolism-related targets at each developmental stage of MASLD and consolidates targeted treatments and therapies, encompassing preclinical evaluations and clinical trials. Currently, targeted drug therapy for liver diseases emphasizes the identification of novel targets and biomarkers, strategies to address drug resistance, innovative therapeutic technologies, and personalized drug applications, as the majority of MASLD-related medications exhibit inconsistent efficacy and safety and have not been extensively developed for patient utilization. Investigating disease-associated signaling networks, including Wnt/β-catenin, Hedgehog, and Notch, can reveal novel targets and guide the development of tailored therapies for liver illnesses. Non-coding RNA targets, such as microRNAs and lncRNAs, also influence the progression of liver disease. Following the announcement of the 24th Nobel Prize in Physiology or Medicine, microRNA gained prominence and is anticipated to have a more significant role in MASLD. To enhance therapeutic efficacy and address drug resistance, multi-target combination therapies can be formulated and refined. Innovative small-molecule delivery technologies such as ADCs, nanotechnology-based methods, and exosomal delivery systems can be employed to specifically target the liver. We will enhance pharmaceutical safety and resistance research, the clinical monitoring and management of targeted therapies, the prompt detection and management of adverse drug reactions, and patient safety during medication administration. Integrating pharmacological agents with alternative mechanisms, formulating novel targeted therapies, or employing intermittent drug administration to mitigate or circumvent drug resistance can enhance the sustained efficacy of targeted medications. Numerous pharmaceuticals function synergistically to enhance therapeutic outcomes. Ultimately, personalized treatment for patients involves analyzing genetic, proteomic, and metabolic data to identify disease biomarkers and formulate tailored treatment strategies; this necessitates the creation of a precision medicine platform that integrates multi-omics data (such as genomics, transcriptomics, proteomics, and metabolomics) with clinical information. Big data analysis and artificial intelligence will be employed to meticulously study patients’ disease characteristics, facilitating personalized treatment. The precision medicine platform enhances patient stratification, treatment effectiveness, and quality of life.

## 8. Limitations

This review has a few restrictions. Due to space limits, most of this review was removed during secondary editing. For example, the majority of the document only discusses highly recommended drugs that have been verified in clinical trials in recent years, whereas drug studies on preclinical compounds are incompletely reported, and fewer natural or Chinese medicines are summarized and numbered. Furthermore, the keyword combinations of targets or targeted drugs used in the literature search, while carefully designed, may have been omitted, resulting in the failure to retrieve some relevant studies, or this review may not have covered the most recent preclinical or early clinical trial data due to the rapid advancement of research in this field and the emergence of new drugs and therapeutic strategies.

## Figures and Tables

**Figure 1 ijms-26-04077-f001:**
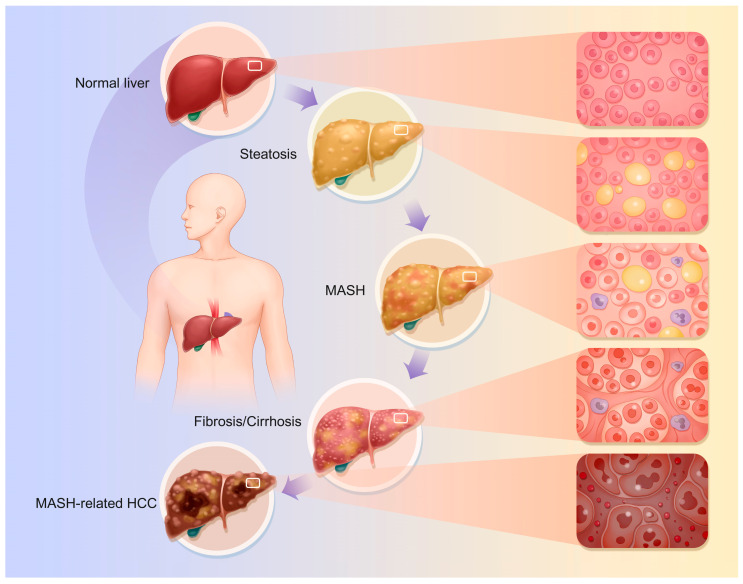
Pathophysiological changes of MASLD. The continuum of disease progression in MASLD encompasses the transition from a normal liver to steatosis, then to MASH, followed by fibrosis and cirrhosis, ultimately culminating in MASH-related HCC. The progression of disease in MASLD is characterized by a transition from normal liver to steatosis, followed by MASH, fibrosis, cirrhosis, and ultimately MASH-related HCC. Nonetheless, as previously indicated, this does not represent a singular linear progression relationship.

**Figure 2 ijms-26-04077-f002:**
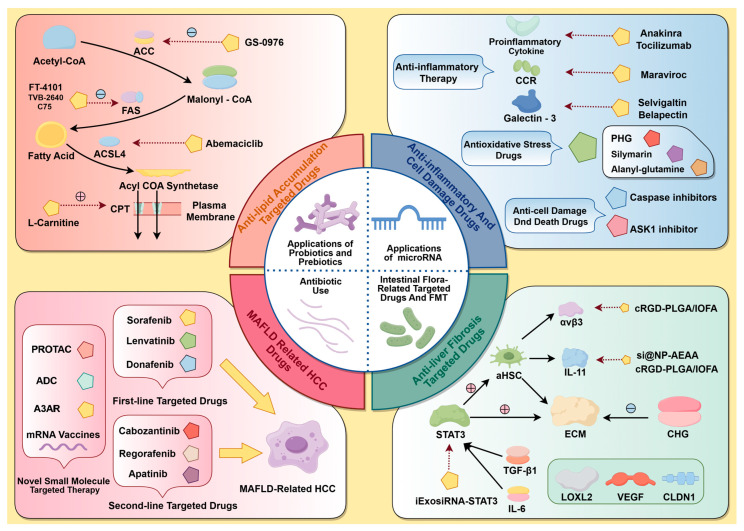
A summary of MASLD-associated targets and their respective targeted therapies. Topics covered include steatosis, MASH, fibrosis, cirrhosis, MASH-related HCC, and the influence of gut microbiota and miRNAs on metabolism and disease progression.

**Table 1 ijms-26-04077-t001:** Clinically trialed lipid-metabolizing drugs.

Objective	Drug	Mechanism	Registered Clinical Trails	Outcome	Duration	Ref.
ACC	PF-05221304	ACC1/2 inhibitor	Phase 2a (NCT03248882)	(↓) Liver fat (↑) Plasma TG	16 weeks	[115]
	MK-4074	ACC1/2 inhibitor	Phase 1 (NCT01431521)	(↓) Liver TG (↑) Plasma TG	1 month	[116]
	NDI-010976	ACC1/2 inhibitor	Phase 1 (NCT02876796)	(↓) Liver DNL	1 week	[117]
	GS-0976	ACC inhibitor	Phase 2 (NCT02856555)	(↓) LFC (↓) MRI-PDFF (↓) Steatosis (↓) Liver injury markers (↑) Plasma TG	12 weeks	[118]
FAS	FT-4101	FAS inhibitor	Phase 1/2 (NCT04004325)	(↓) Liver DNL	12 weeks	[119]
	TVB-2640	FAS inhibitor	Phase 2 (NCT03938246)	(↓) Liver fat (↓) Hepatitis markers	12 weeks	[120]
			Phase 2 (NCT04906421)	(↓) Liver fat (↓) Hepatitis	52 weeks	[121]
ACLY	Bempedoic acid	ACLY inhibitor	Phase 3 (NCT02666664)	(↓) LDL-C	12 weeks	[122]
HMG-CoA reductase	Statins	HMG-CoA reductase inhibitor	Phase 2 (NCT02633956)	(↓) LDL-C	4 weeks	[123]
			Phase 2/3 (NCT03758058)	(↓) ALT (↓) AST	10 weeks	[124]
SCD1	Aramchol	SCD1 inhibitor	Phase 2b (NCT02279524)	(↓) Liver fat (↓) Hepatitis	52 weeks	[125]
			Phase 1b/2a (NCT04140123)	(↓) LFC (↓) Liver fibrosis	28 days	[126]
DGAT2	IONIS-DGAT2Rx	Antisense oligonucleotide inhibitor	Phase 2 (NCT03334214)	(↓) Liver fat	13 weeks	[127]
L–THR	Resmetirom	THR-β agonist	Phase 3 (NCT03900429)	(↓) LDL-C (↓) Hepatitis	52 weeks	[128]
			Phase 3 (NCT04197479)	(↓) LDL-C (↓) Apo B (↓) TG (↓) Liver fat	52 weeks	[129]
PDE	ZSP1601	Inhibition of PDE	Phase Ib/IIa (NCT04140123)	(↓) ALT (↓) AST (↓) LFC	28 days	[126]
	PTX	Inhibition of PDE	Phase 2 (NCT00590161)	(↓) Liver fibrosis (↓) ALT (↓) Steatohepatitis (↓) Hepatic steatosis	1 year	[130]
SGLT2	Empagliflozin	SGLT2 inhibitor	Phase 4 (NCT04642261)	(↓) Liver fat (↓) ALT (↓) MRI-PDFF	52 weeks	[131]
	Dapagliflozin	SGLT2 inhibitor	Phase 1 (ChiCTR2100054612)	(↓) CAP (↓) LAM (↓) LFC (↓) ALT (↓) TNF-α (↓) IL-6	24 weeks	[132]
	Ipragliflozin	SGLT2 inhibitor	Phase 2/3 (UMIN000015727)	(↓) HbA1c (↓) BMI (↓) Liver fibrosis	24 weeks	[133]
			Phase 3 (UMIN000022651)	(↓) AST (↓) ALT (↓) HbA1c (↓) FBG (↓) Body weight (↓) Visceral fat area	24 weeks	[134]
			Phase 3 (PMID34558835)	(↓) HbA1c (↓) BMI	72 weeks	[133]
	Canagliflozin	SGLT2 inhibitor	Phase 3 (NCT02065791)	(↓) HbA1c (↓) Weight (↓) ALT (↓) AST (↓) GGT (↓) AP (↑) Bilirubin	2.62 years	[135]
PPAR	Pemafibrate	Selective PPAR-α modulator	Phase 2 (NCT03350165)	(↓) ALT (↓) LDL-C (↓) Liver stiffness	72 weeks	[136]
			Phase 3 (NCT03071692)	(↓) TG (↓) VLDL (↓) Cholesterol (↓) Apo C-III	3.75 years	[137]
	Pioglitazone	PPAR-γ sensitizer	Phase 4 (NCT00994682)	(↓) ALT (↓) AST (↓) GGT (↓) HOMA-IR (↓) ADIPO-IR (↓) VAI	12 months	[138]
	MSDC-0602K	PPAR-γ sensitizer	Phase 2b (NCT 02784444)	(↓) insulin (↓) glycosylated hemoglobin (↓) ALT (↓) AST	52 weeks	[139]
	Lanifibranor	pan-PPAR agonist	Phase 2b (NCT03008070)	(↓) AST (↓) Lipids (↓) Inflammation (↓) Liver fibrosis	24 weeks	[140]
	Saroglitazar	PPAR-α/γ agonist	Phase 2 (NCT03061721)	(↓) Total Cholesterol (↓) TG (↓) LDL-C (↓) VLDL-C	6 weeks	[141]
			Phase 3 (NCT03863574)	(↓) Total Cholesterol (↓) TG (↓) LDL-C (↓) VLDL-C (↓) Blood lipid	24 weeks	[142]
	Elafibranor (GFT505)	PPAR-α/δ agonist	Phase 2 (NCT01694849)	(↓) Liver fibrosis	52 weeks	[143]
			Phase 2 (NCT03883607)	(↓) ALT (↓) AST (↓) Liver fibrosis (↓) Inflammatory	12 weeks	[144]
GLP-1	Semaglutide	GLP-1RA	Phase 2 (NCT02970942)	(↓) ALT (↓) AST (↓) Weight (↓) Fasting blood sugar (↓) Liver fibrosis	72 weeks	[145]
	Efinopegdutide	GLP-1/ GCG dual receptor agonist	Phase 2a (NCT04944992)	(↓) ALT (↓) AST (↓) Weight (↓) Fasting blood sugar (↓) Liver fibrosis	12 months	[146]
	Liraglutide	GLP-1 analogue	Phase 2 (NCT01237119)	(↓) Intrahepatic fat (↓) Abdominal fat (↓) Body weight	26 weeks	[147]
	Cotadutide	GLP-1/glucagon receptor dual agonist	Phase 2a (NCT03555994)	(↓) Liver glycogen (↓) Liver fat	28 days	[148]
	Dulaglutide	Glucagon-like peptide-1 receptor agonist	Phase 2 (NCT03590626)	(↓) LFC (↓) Liver stiffness (↓) GGT (↓) AST (↓) ALT	24 weeks	[149]
	Tirzepatide (LY3298176)	GLP-1R/GIPR dual agonist	Phase 2b (NCT04166773)	(↓) ALT (↓) AST (↓) Weight (↓) Liver fibrosis (↓) HbA1c	52 weeks	[150]

**Table 2 ijms-26-04077-t002:** Anti-inflammatory and anti-cell-death drugs in clinical trials.

Target	Drug	Mechanism	Registered Clinical Trails	Outcome	Duration	Ref.
CCR	CCV	Bifunctional CCR antagonist	Phase 2b (NCT02217475)	(↓) Liver fibrosis	24 months	[214]
FXR	OCA	Selective FXR agonist	Phase 3 (NCT02548351)	(↓) ALT (↓) AST (↓) Liver fibrosis	52 weeks	[215]
	Tropifexor	FXR agonist	Phase 2 (NCT03517540)	(↓) ALT (↓) AST (↓) Liver fibrosis (↓) Weight	48 weeks	[216]
	Cilofexor (GS-9674)	FXR agonist	Phase 2 (NCT02854605)	(↓) MRI-PDFF (↓) GGT (↓) Primary BAs	24 weeks	[217]
	HTD1801 (BUDCA)	FXR agonist	Phase 2 (NCT03656744)	(↓) LFC (↓) Liver-related Enzymes (↓) Body weight (↓) Blood glucose	18 weeks	[218]
FGFs	Pegbelfermin (BMS-986036)	FGF21 agonist	Phase 2b (NCT03486899, NCT03486912)	(↓) Hepatic steatosis (↓) Hepatic fibrosis (↓) MRI-PDFF (↓) AST (↓) ALT	48 weeks	[219,220]
	Efruxifermin	Long-lasting Fc-FGF21 fusion protein	Phase 2a (NCT03976401)	(↓) Liver fat (↓) Liver injury (↓) Liver fibrosis	16 weeks	[221]
			Phase 2a (NCT04767529)	(↓) Liver fibrosis (↓) HFF	96 weeks	[221]
	Aldafermin	FGF19 analogue	Phase 2 (NCT02443116)	(↓) Liver fat (↓) CT1 (↓) ALT (↓) AST (↓) Liver fibrosis	12 weeks	[222]
			Phase 2b (NCT03912532)	(↓) Fatty liver (↓) Inflammation (↓) Liver injury (↓) Liver fibrosis	24 weeks	[223]
ROS	Vitamin E	Fat-soluble antioxidant	Phase 3 (NCT00063622)	(↓) NASH (↓) AST (↓) ALT (↓) Hepatitis severity	96 weeks	[224]
	ALA	Antioxidant	IRCT201511143320N12	(↓) Liver fat	12 weeks	[225]
ASK1	Selonsertib	ASK1 inhibitor	Phase 2 (NCT02466516)	(↓) Cirrhosis (↓) Liver fat (↓) Biomarkers of necrosis	24 weeks	[226]
			Phase 3 (NCT03053050, NCT03053063)	Minimal liver fibrosis decrease	48 weeks	[227]

**Table 3 ijms-26-04077-t003:** Targeted cutting-edge experiments for the treatment of liver fibrosis.

Target	Drugs	Mechanism	Outcome	Ref.
HSC/ECM	BMS-986263 Phase 2 (NCT03420768)	Silencing of HSP47 mRNA by SiRNA delivered by lipid nanoparticles	(↓) HSP47 mRNA (↓) HSP47 protein (↓) ECM (↓) Liver fibrosis	[263]
	AA-T3A-C12/siHSP47 LNP	Silencing of HSP47 mRNA by SiRNA delivered by lipid nanoparticles	(↓) HSP47 mRNA (↓) HSP47 protein (↓) ECM (↓) Liver fibrosis	[264]
	HCQ@ROL-LNPs	Targeted inhibition of autophagy in aHSCs	(↓) Autophagy in aHSC (↓) ECM (↓) Liver fibrosis	[265]
	cRGD-PLGA/IOFA NPs	Co-targeting macrophages and HSCs	(↓) ECM (↓) Liver fibrosis	[266]
	si@NP-AEAA	IL-11/ERK siRNA blockage inhibits aHSC production	(↓) ECM (↓) Liver fibrosis (↓) Hepatitis	[267]
	Polymer containing miR-155-resistant CXCR4 inhibitor	Preventing HSC-KC crosstalk	(↓) ECM (↓) Liver fibrosis	[268]
	Novel fluoropolymers efficiently administer anti-miR155	Concurrent regulation of KCs and HSCs	(↓) ECM (↓) Liver fibrosis (↓) Hepatitis	[269]
	COL-HA-GA	Collagen degradation and HSC inhibition	(↓) ECM (↓) aHSCs (↓) Liver fibrosis	[270]
	COL + SLB-MLPs	Collagen degradation and HSC inhibition	(↓) ECM (↓) aHSCs (↓) Liver fibrosis	[271]
	IGNP-JQ1	Reopening of LSEC to inhibit HSC activation	(↓) Porosity of LSECs (↓) ECM (↓) aHSCs (↓) Liver fibrosis	[272]

**Table 4 ijms-26-04077-t004:** HCC confirmatory and emerging therapies.

Class	Drug	Major Mechanism	Registered Clinical Trails	Evaluation of Efficacy	Ref.
Clinically used drugs	Sorafenib	Targeting RAF kinase, VEGFR, PDGFR, KIT, and FLT3	FDA-approved (2007)	As first-line medication	[293,294]
	Regorafenib	Targeting VEGFR-2, VEGFR-3, KIT, RET, RAF-1, BRAF, PDGFR, and FGFR	FDA-approved (2017)	As second-line medication	[295]
	Lenvatinib	Targeting VEGFR, FGFR, PDGFRα, KIT, and RET	FDA-approved (2018)	As first-line medication	[296]
	Pembrolizumab	Targeting PD-1	FDA-approved (2018)	As second-line medication	[297]
	Cabozantinib	Targeting MET, VEGFR, ROS1, RET, AXL, NTRK, and KIT	FDA-approved (2019)	As second-line medication	[298,299]
	Ramucirumab	Targeting VEGFR2	FDA-approved (2019)	As second-line medication	[300,301]
	Atezolizumab plus bevacizumab	Targeting PDL1, and VEGFA	FDA-approved (2020)	As first-line medication	[302,303]
Novel small-molecule therapies	XZ1606 (LIVTAC)	PROTAC molecules degrade BRD4	Preclinical experiment	(↓) BRD4, c-Myc (↓) HCC	[304]
	GPC3-specific ADC	Dedicated ADC for GPC3 degradation	Preclinical experiment	Showing HCC’s cytotoxicity	[305]
	CLDN6-specific ADC plus DM1	Specific ADC-targeted CLDN6 degradation and DM1 cytotoxicity	Preclinical experiment	Showing HCC’s cytotoxicity	[306]
	Namodenoson (CF102)	A3AR agonism modulates Wnt and NF-kB signaling, promoting HCC cell death	Clinical experiment Phase 1/2	(↑) Overall survival in patients with HCC (↑) Progression-free survival in patients with HCC	[307]

## Data Availability

Not applicable.

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
