# Peer review of "Targeting Metabolism: Innovative Therapies for MASLD Unveiled"

_ijms, 2025, doi:10.3390/ijms26094077_

Round 1

Reviewer 1 Report

Comments and Suggestions for Authors

The manuscript "Targeting Metabolism: Innovative Therapies for MASLD 2 Unveiled" authored by Wang et al. aims to discuss the reclassification of MASLD and its implications for diagnosis, treatment, and metabolic research. However, the manuscript is filled with significant issues related to clarity and accuracy, of writing, which often reflects a superficial understanding of the topic, characterized by vague statements, redundant explanations, and misleading descriptions. These issues are evident from the very beginning, as even the first sentence provides incorrect information. Specifically, the term "metabolic syndrome" is singular (not plural) and refers to a number of conditions that increase the risk of e.g. cardiovascular diseases, and diabetes. By this said, obesity and type 2 diabetes mellitus are components of metabolic syndrome, not separate "metabolic syndromes" as stated by the authors.

The major problem with the manuscript is that misleading statements occur frequently throughout the manuscript. For example, one of more advanced stages of MASLD are "steatohepatitis", not just "hepatitis" (line 32). Next, the phrase ‘malignant transformations’ (line 32) is vague and potentially misleading, as it implies a general malignant change rather than specifying the increased risk of hepatocellular carcinoma (HCC). Additionally, the term ‘secondary liver cancer’ is ambiguous. If referring to metastases from other primary tumors, it should be clarified; if hepatocellular carcinoma is intended, ‘primary liver cancer’ is the appropriate term. Furthermore, refferal to portal hypertension is unclear.

Another issue arises with the statement, “The name MASLD offers a more precise characterization,” which is misleading, as MASLD is a reclassification rather than a newly introduced concept. The phrase, “Besides fatty liver (steatosis) in NAFLD, MASLD encompasses...,” is problematic because it implies that NAFLD only includes steatosis, whereas it also covered NASH (now MASH) and associated complications. Next, the transition “It includes a wider range of pathology” is inaccurate: MASLD does not necessarily include a broader range of pathology than NAFLD, but rather redefines the condition based on metabolic dysfunction. Such mistakes highlight the complete lack of the knowledge in the field of MASLD. 

The sentence, “The new definition highlights a paradigm change in understanding metabolic anomalies as a causal factor” (lines 103-104), is misleading. While MASLD emphasizes metabolic dysfunction, it does not introduce metabolic causes as a new concept: NAFLD has long been associated with metabolic dysfunction.

Furthermore, the manuscript is plagued by repeated sentences, thus paragraphs lack the logical flow of the argument and depth. For example, the discussion of metabolic abnormalities is unnecessarily repeated (lines 57 and 61), creating the impression that the authors are filling the space in the paragraphs in the manuscript rather than offering any novel insights. Similarly, the impact of the new MASLD definition on early identification and screening is repeated (lines 95-102) within the same paragraph.

The manuscript also fails in terms of introducing its content, for example, Table 1 appears before it is mentioned in the text, disrupting the logical flow and confusing the reader. Additionally, the introduction of the terms of hepatic steatosis and MASH does not align with the well-established definitions. These inaccuracies undermine the credibility of the manuscript and its claims.

In summary, I have the impression there is no single page in the manuscript that does not require substantial correction. The frequent errors, redundancies, and lack of clarity throughout the document are significant enough to warrant a recommendation for rejection. Without the general rewriting to address these issues, the manuscript does not meet the standards expected for scientific rigor and clarity.

Author Response

Comments 1: The term "metabolic syndrome" is singular (not plural) and obesity and type 2 diabetes mellitus are components of metabolic syndrome, not separate "metabolic syndromes" as stated by the authors.

Response 1: Thank you for pointing this out, we agree with this comment. We have amended the description of "metabolic syndrome" and revised the sentence to "The increasing global prevalence of metabolic syndrome, including obesity and type 2 diabetes mellitus (T2DM), is significantly contributing to the rise in chronic hepatic steatosis associated with these metabolic disorders". It can be found on line 28 of the revised version.

Comments 2: One of more advanced stages of MASLD are "steatohepatitis", not just "hepatitis" (line 32). 

Response 2: Thank you for pointing this out, we agree with this comment. We have amended "hepatitis" there to "steatohepatitis", which can be found in line 33 of the revised version. And we have carefully reviewed the entire manuscript to make sure that there are no similar issues.

Comments 3: The phrase ‘malignant transformations’ (line 32) is vague and potentially misleading, as it implies a general malignant change rather than specifying the increased risk of hepatocellular carcinoma (HCC).

Response 3: Thank you for pointing this out, we agree with this comment. We've changed "malignant transformations" to "malignant injury resulting in liver function impairment and the development of hepatocellular carcinoma (HCC) ", it can be found in the revised version on line 34.

Comments 4: The term ‘secondary liver cancer’ is ambiguous. If referring to metastases from other primary tumors, it should be clarified; if hepatocellular carcinoma is intended, ‘primary liver cancer’ is the appropriate term.

Response 4: Thank you for pointing this out, we agree with this comment. We've changed "secondary liver cancer" to "primary malignant HCC", which can be found on line 36 of the revised version.

Comments 5: The relationship to portal hypertension is unclear.

Response 5: Thank you for pointing this out, we agree with this comment. We've deleted “complications related to portal hypertension (e.g., rupture and bleeding of esophagogastric varices, ascites, hepatic encephalopathy),” to minimize the appearance of misleading language.

Comments 6: Another issue arises with the statement, "The name MASLD offers a more precise characterization, " which is misleading, as MASLD is a reclassification rather than a newly introduced concept. 

Response 6: Thank you for pointing this out, we agree with this comment. We've modified the sentence to" The name MASLD provides a reclassification of fat-related liver disease", it can be found on line 96 of the revised version. Furthermore, we reviewed the full manuscript and revised the description of MASLD as a newly introduced concept, e.g., changing "The New Definition: MASLD" to "The New Classification: MASLD", "new definition" to "reclassified term", and so on.

Comments 7: The phrase, “Besides fatty liver (steatosis) in NAFLD, MASLD encompasses...,” is problematic because it implies that NAFLD only includes steatosis, whereas it also covered NASH (now MASH) and associated complications.

Response 7: Thank you for pointing this out, we agree with this comment. We've modified the sentence to "It reclassifies a range of diseases according to metabolic dysfunction: steatosis, metabolic dysfunction-associated steatohepatitis (MASH), fibrosis, cirrhosis, and MASH-related HCC", removing the implication and misinterpretation in the description that "NAFLD only includes steatosis". It can be found in line 98 of the revised version.

Comments 8: Next, the transition "It includes a wider range of pathology" is inaccurate: MASLD does not necessarily include a broader range of pathology than NAFLD, but rather redefines the condition based on metabolic dysfunction. 

Response 8: Thank you for pointing this out, we agree with this comment. We've deleted "It includes a wider range of pathology", reducing narrative misunderstandings.

Comments 9: The sentence, “The new definition highlights a paradigm change in understanding metabolic anomalies as a causal factor” (lines 103-104), is misleading. While MASLD emphasizes metabolic dysfunction, it does not introduce metabolic causes as a new concept: NAFLD has long been associated with metabolic dysfunction.

Response 9: Thank you for pointing this out, we agree with this comment. We've modified the sentence to "MASLD recategorizes diseases according to metabolic disorders", reducing narrative misdirection. It can be found on line 108 of the revised version.

Comments 10: Furthermore, the manuscript is plagued by repeated sentences, thus paragraphs lack the logical flow of the argument and depth. For example, the discussion of metabolic abnormalities is unnecessarily repeated (lines 57 and 61)。

Response 10: Thank you for pointing this out, we agree with this comment. We removed the redundant sentence "Lastly, it provides an overview and analysis of these specific therapeutic drugs for the treatment of MASLD. Along with reviewing the state-of-the-art targeted medications (including those in preclinical and clinical studies), the review also looks at the recent metabolic abnormalities linked to each stage of MASLD", ensuring a concise flow of the manuscript.

Comments 11: Similarly, the impact of the new MASLD definition on early identification and screening is repeated (lines 95-102) within the same paragraph.

Response 11: Thank you for pointing this out, we agree with this comment. We've modified the sentence to “The reclassified definition broadens the parameters of screening and enables the early identification of a greater number of individuals at risk for metabolic disorders, including those who are overweight or obese, those diagnosed with type 2 diabetes, and other high-risk populations. Individuals may be assessed for MASLD using fundamental liver function tests and ultrasound imaging, even in the absence of overt liver disease manifestations”, combining redundant sentences into concise descriptions. It can be found on line 102 of the revised version.

Comments 12: Table 1 appears before it is mentioned in the text, disrupting the logical flow and confusing the reader.

Response 12: Thank you for pointing this out, we agree with this comment. We have placed all the Table and Figure locations closest to the citation to ensure a smooth and logical read.

Comments 13: Additionally, the introduction of the terms of hepatic steatosis and MASH does not align with the well-established definitions. 

Response 13: Thank you for pointing this out, we agree with this comment. We consulted relevant sources, such as EASL -EASD-EASO Clinical Practice Guidelines on the Management of Metabolic Dysfunction-Associated Steatotic Liver Disease (MASLD), relevant amendments were added to the manuscript, for example, "MASH is characterized by histological features of hepatocellular ballooning and lobular inflammation… " ,"Hepatic steatosis is the hallmark of MASLD…","MASLD is characterized by excess triglyceride accumulation in the liver alongside at least one cardiometabolic risk factor”.

In summary, according to your relevant comments, we have revised the content of the relevant specific comments, and carefully reviewed and corrected each page of the entire manuscript to ensure that the issues you raised have been comprehensively repaired, so that the manuscript's correctness, conciseness, logic, readability and other aspects of the manuscript to improve, and we hope that the revised manuscript will be able to reach the sufficient scientific rigor.

Reviewer 2 Report

Comments and Suggestions for Authors

The manuscript offers an in-depth review of metabolic dysfunction-associated steatotic liver disease (MASLD), highlighting metabolism's vital role in its pathophysiology. In my opinion, this subject is particularly significant considering the rising global prevalence of MASLD. However, I found multiple areas needing revision that would enhance the manuscript's scientific quality and impact.

General comments:

1. The manuscript does not reference the entity "MetALD" (metabolic and alcohol-associated steatotic liver disease), which should be included in a thorough review of this field.

2. This manuscript makes a major contribution by including recent clinical trial data for agents such as efruxifermin, pegbelfermin, pemafibrate, resmetirom, and semaglutide. However, the authors should also consider referencing other relevant clinical trials targeting metabolism in MASLD, such as NCT03970031 and NCT05284448, which focus on an insulin sensitizer and phosphodiesterase inhibitors, respectively.

3. Targeting metabolism as an innovative therapy for MASLD is a positive aspect of the manuscript. However, several sections towards the end, such as microRNA, mRNA vaccines, and nanosensors in HCC, do not directly relate to this focus. These sections may distract the reader from the main topic and should be removed or significantly condensed to maintain the manuscript's coherence and focus.

4. The revision does not sufficiently describe the non-linear nature of the MASLD disease process. For example, some patients exhibit fibrosis without developing NASH, and some others may even reverse the degree of severity from MASH to steatosis. This concept regarding "non-linear nature" should be stated in the text and accurately represented in Figure 1, as the current figure may suggest a linear disease progression.

Specific comments:

5. Regarding Figure 1, I suggest using actual histological images instead of drawings to provide a more accurate representation of the histological changes occurring in this disease.

6. In line 895, this sentence should be modified to "In liver fibrosis patients, capillarized sinusoids." In addition, "endothelial fenestration" is more accurate than "endothelial opening," as mentioned in line 905.

7. In line 877, "Collagen degradation and HSC inhibition" should be corrected to "Collagen degradation and inhibition of HSC activation."

8. In line 202, the sentence "Advanced livers are smaller and uneven" should be corrected to "Advanced fibrosis (or advanced fibrotic livers) are smaller and uneven."

9. In line 472, the authors should clarify the sentence "regarding the method that relies on normal mitochondria." It is not clear to me what it means, and is redundant with the following line.

Author Response

Comments 1: The manuscript does not reference the entity "MetALD" (metabolic and alcohol-associated steatotic liver disease), which should be included in a thorough review of this field.

Response 1: Thank you for pointing this out, we agree with this comment. We have therefore added a description of MetALD to the manuscript, which mainly includes “Notably, in addition to MASLD, the management consensus states that SLD also comprises MASLD with moderate alcohol intake (MetALD)……The diagnosis of MASLD or MetALD is based on whether alcohol is consumed or not, as well as if alcohol consumption is excessive (≥20 g/day for women and ≥30 g/day for males)”,it can be found on line 84 of the revised version.

Comments 2: This manuscript makes a major contribution by including recent clinical trial data for agents such as efruxifermin, pegbelfermin, pemafibrate, resmetirom, and semaglutide. However, the authors should also consider referencing other relevant clinical trials targeting metabolism in MASLD, such as NCT03970031 and NCT05284448, which focus on an insulin sensitizer and phosphodiesterase inhibitors, respectively.

Response 2: Thank you for pointing this out, we agree with this comment. We've added relevant new content that you've pointed out, such as "And MSDC-0602K is a second-generation thiazolidinedione that…","Phosphodiesterase (PDE) is an enzyme family…", which can be found in lines 595 and 503 of the revised version, respectively.

Comments 3: Targeting metabolism as an innovative therapy for MASLD is a positive aspect of the manuscript. However, several sections towards the end, such as microRNA, mRNA vaccines, and nanosensors in HCC, do not directly relate to this focus. These sections may distract the reader from the main topic and should be removed or significantly condensed to maintain the manuscript's coherence and focus.

Response 3: Thank you for pointing this out, we agree with this comment. We have significantly condensed or deleted the content you described above to increase the coherence of the manuscript.

Comments 4: The revision does not sufficiently describe the non-linear nature of the MASLD disease process. For example, some patients exhibit fibrosis without developing NASH, and some others may even reverse the degree of severity from MASH to steatosis. This concept regarding "non-linear nature" should be stated in the text and accurately represented in Figure 1, as the current figure may suggest a linear disease progression.

Response 4: Thank you for pointing this out, we agree with this comment. Therefore, we have added a description of "It is worth noting, this spectrum represents a nonlinear relationship; specifically, while MASLD suggests a theoretical progression of exacerbation, some patients may display fibrosis without progressing to MASH, and others may even revert from severe MASH to steatosis" in the "Pathological Changes in MASLD" section, which you can see on line 150 of the revised version; And we've added pertinent description of it in Figure 1, which you can see in line 158 of the revised version.

Comments 5: Regarding Figure 1, I suggest using actual histological images instead of drawings to provide a more accurate representation of the histological changes occurring in this disease.

Response 5: Thank you for your suggestion, but after some discussion and searching, we found it difficult to find images of actual tissues that were particularly typical, or whose features were not typical, so we believe that the current form of drawing is more likely to allow the reader to capture the relevant focus of histology and better understand it, in order to characterize the phases described in this review.

Comments 6: In line 895, this sentence should be modified to "In liver fibrosis patients, capillarized sinusoids." In addition, "endothelial fenestration" is more accurate than "endothelial opening," as mentioned in line 905.

Response 6: Thank you for pointing this out, we agree with this comment. We've modified the sentence to “In liver fibrosis patients, capillarized sinusoids…”, which can be found in line 931 of the revised version; and for the next change we also changed the term to "endothelial fenestration", which can be found in line 939 of the revised version.

Comments 7: In line 877, "Collagen degradation and HSC inhibition" should be corrected to "Collagen degradation and inhibition of HSC activation."

Response 7: Thank you for pointing this out, we agree with this comment. We've modified the sentence to “Collagen degradation and inhibition of HSC activation.”, which can be found in line 912 of the revised version.

Comments 8: In line 202, the sentence "Advanced livers are smaller and uneven" should be corrected to "Advanced fibrosis (or advanced fibrotic livers) are smaller and uneven."

Response 8: Thank you for pointing this out, we agree with this comment. We've modified the sentence to “Advanced fibrotic livers are smaller and uneven.”, which can be found at line 207 of the revised version.

Comments 9: In line 472, the authors should clarify the sentence "regarding the method that relies on normal mitochondria." It is not clear to me what it means, and is redundant with the following line.

Response 9: Thank you for pointing this out, we agree with this comment. This sentence was redundant and caused you to misunderstand it, so we've changed the sentence to "The method via which resmetirom lowers hepatic fat in people with MASLD may rely on the reestablishment of normal mitochondrial function and enhanced beta-oxidation”, making sentences flow better and less likely to cause misunderstandings. It can be found in line 481 of the revised version.

Round 2

Reviewer 2 Report

Comments and Suggestions for Authors

The author's responses address most of my questions.